



# Filling the gaps of in-situ hourly PM₂.₅ concentration data with the aid of empirical orthogonal function constrained by diurnal cycles

Kaixu Bai[1,2], Ke Li[2], Jianping Guo[3], Yuanjian Yang[4,5], Ni-Bin Chang[6]

[1]Key Laboratory of Geographic Information Science (Ministry of Education), East China Normal University, Shanghai 200241, China

[2]School of Geographic Sciences, East China Normal University, Shanghai 200241, China

[3]State Key Laboratory of Severe Weather, Chinese Academy of Meteorological Sciences, Beijing 100081, China

[4]School of Atmospheric Physics, Nanjing University of Information Science & Technology, Nanjing, China

[5]Institute of Environment, Energy and Sustainability, The Chinese University of Hong Kong, Hong Kong, China

[6]Department of Civil, Environmental, and Construction Engineering, University of Central Florida, Orlando, FL 32816, USA

*Correspondence to*: Dr./Prof. Jianping Guo (jpguocams@gmail.com)



**Abstract.** Data gaps are frequently observed in the hourly $PM_{2.5}$ mass concentration records measured from the China national air quality monitoring network. In this study, we proposed a novel gap filling method called the diurnal cycle constrained empirical orthogonal function (DCCEOF) to fill in data gaps present in hourly $PM_{2.5}$ concentration records. This method mainly calibrates the diurnal cycle of $PM_{2.5}$

that is reconstructed from discrete $PM_{2.5}$ neighborhood fields in space and time to the level of valid $PM_{2.5}$ data values observed at adjacent times. Prior to gap filling, possible impacts of varied number of data gaps in the time series of hourly $PM_{2.5}$ concentration on $PM_{2.5}$ daily averages were examined via sensitivity experiments. The results showed that $PM_{2.5}$ data suffered from the gaps on about 40% of days, indicating a high frequency of missing data in the hourly $PM_{2.5}$ records. These gaps could introduce

significant bias to daily-averaged $PM_{2.5}$. Particularly, given the same number of gaps, larger biases would be introduced to daily-averaged $PM_{2.5}$ during clean days than polluted days. The cross-validation results indicate that the predicted missing values from the DCCEOF method with the consideration of the local diurnal phases of $PM_{2.5}$ are more accurate and reasonable than those from the conventional spline interpolation approach, especially for the reconstruction of daily peaks and/or minima that cannot be

restored by the latter method. To fill the gaps in the hourly $PM_{2.5}$ records across China during 2014 to 2019, as a practical application, the DCCEOF method can be able to reduce the averaged frequency of missingness from 42.6% to 5.7%. In general, the present work implies that the DCCEOF method is realistic and robust to be able to handle the missingness issues in time series of geophysical parameters with significant diurnal variability and can be expectably applied in other data sets with similar barriers

because of its self-consistent capability.



## 1 Introduction

A large variety of ground-based monitoring networks have been established worldwide to provide accurate measurements on various aspects of the atmospheric environment such as the Aerosol Robotic Network (AERONET) for aerosol properties. Many of these in-situ measurements, however, suffer from data losses due to various unexpected accidents, e.g., instrumental malfunction, interruption of power supply, internet outage either on monitoring stations or user's end, thereby resulting in salient data gaps in the archived data records. Undoubtedly, these gaps significantly impair the data qualities and their valuable applications. Therefore, filling the data gaps present in such datasets is critical and of great value to facilitating the broad application of in-situ measurements.

Confronted with frequent severe haze pollution events, China started to establish the national ambient air quality monitoring network since 2012 by extending the range of the previous sparsely distributed monitoring network to cover most major Chinese cities. To date, more than 1,600 state-level stations routinely operate to measure concentrations of six essential air pollutants (i.e., $PM_{10}$, $PM_{2.5}$, $O_3$, $NO_2$, $SO_2$, CO) on an hourly basis (Guo et al., 2017; Li et al., 2017a). These in-situ measurements are publicly released online via the China National Environment Monitoring Centre (CNEMC) in near real-time as of 2013 but without providing any direct data download interface. Consequently, users oftentimes utilize an automated software program (often known as a "web crawler") to retrieve these valuable data sources from the CNEMC website. Such an endeavour helps users to acquire hourly air quality data more efficiently. The retrieved hourly mass concentration record, taking $PM_{2.5}$ for instance, has been widely used as a critical data source in many studies related to haze pollutions, because of its good accuracy and high temporal resolution as well as its national-scale coverage (Gao et al., 2018; Miao et al., 2018; Bai et al., 2019a, 2019b; Zhang et al., 2019).

Although $PM_{2.5}$ data from this dataset have been extensively used in many $PM_{2.5}$-related studies, the method of treating data gaps during the data cleaning processes, particularly for those using daily or monthly averaged $PM_{2.5}$ data (e.g., Miao et al., 2018; Ye et al., 2018; Zhang et al., 2018; Yang et al., 2019a), is oftentimes unclear. Since ignoring missing values would undoubtedly introduce biases into the final results (Bondon, 2005; Larose et al., 2019), some studies attempted to perform data analysis on a relatively long time scale to mitigate the impacts of data gaps by integrating hourly records into monthly



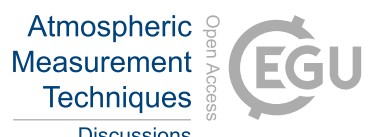

resolution  (e.g., Bai et al., 2019b; Zhang et al., 2019). On the other hand, many previous studies preferred

to exclude records of days with a certain degree of missing values (e.g., no more than 6 missing values

within 24-h) from their analysis (e.g., van Donkelaar et al., 2016; Li et al., 2017; Huang et al., 2018;

Manning et al., 2018; Shen et al., 2018; Bai et al., 2019a; Zhang et al., 2019). Although the exclusion of

records with missingness could avoid biased results to some extent, such a data treatment would make

the $PM_{2.5}$ time series temporally discontinuous. Therefore, approaches of ignoring missing values or

excluding records on days with missing values are unreasonable.

Since a non-gap $PM_{2.5}$ record is essential to $PM_{2.5}$ related haze control and environmental health risk

assessment, filling data gaps presented in hourly $PM_{2.5}$ records are of critical importance. Although

versatile gap filling methods exist (e.g., Beckers and Rixen, 2003; Taylor et al., 2013; Chang et al., 2015;

Dray and Josse, 2015; Gerber et al., 2018), most of them fail to properly impute missingness in $PM_{2.5}$

time series with high temporal resolution (e.g., hourly). An overview of existing gap filling methods is

therefore worthwhile. Some conventional methods working in a principle of statistical interpolation are

incapable of restoring daily peaks and/or minima since a priori knowledge of diurnal phases is oftentimes

required to cope with this issue. The primary reason lies in the varied diurnal phases of $PM_{2.5}$

measurements since the mass concentrations always vary significantly in space and time due to

heterogeneous local emissions and atmospheric conditions (Guo et al., 2017; Lennartson et al., 2018; Shi

et al., 2018). A similar barrier applies for many other datasets which are sampled at high temporal

resolution.

In this study, we proposed a novel practical gap filling method called a diurnal cycle constrained empirical

orthogonal function (DCCEOF) to better handle data gaps presented in time series with marked variability

in space and time, by taking diurnal phases as a critical constraint in missing value imputation. To our

knowledge, none of the existing gap filling methods have accounted for the diurnal phase effect in their

missing value imputation schemes, and hence the predicted values from these methods might suffer from

large bias. As a demonstration, the retrieved hourly $PM_{2.5}$ concentration record from CNEMC during the

time period of 2014 to 2019 was applied to evaluate the efficacy and accuracy of the proposed DCCEOF

method. Science questions to be answered by this study include: (1) how many and how often are the

missing values presented in a large-scale monitoring network such as the one in China with abundant in



situ PM$_{2.5}$ records? (2) what are the uncertainties that can be introduced by missing values to daily averaged PM$_{2.5}$? (3) is it feasible to reconstruct a set of spatiotemporally localized diurnal cycles from

discrete PM$_{2.5}$ observations in a large-scale monitoring network? and (4) are missing value imputations constrained by the diurnal cycles reliable?

## 2 Overview of existing gap filling methods

Plenty of methods have been developed or adopted for gap filling with respect to various theoretical bases, ranging from simple replacement with surrogates to spatial or temporal interpolation in addition to

complicated machine learning techniques such as neural networks. These methods can be classified into different groups according to different criteria. For instance, two major groups can be classified based on the number of variables (univariate versus multivariate) (Ottosen and Kumar, 2019) and theoretical basis (likelihood-based versus imputation-based) (Junger and Ponce de Leon, 2015). Table 1 summarizes a selection of popular methods for missing value imputation in geophysical data sets by referring to the

domain specific data dependence (Gerber et al., 2018). Comparisons of the performances of these methods can also be found in other literatures, e.g., Kandasamy et al. (2013), Demirhan and Renwick (2018), Yadav and Roychoudhury (2018), Julien and Sobrino (2019), among others.

Given that each method is initially proposed to deal with missingness in one specific data set, adopting one method to another data set is often a challenge due to the various features of missingness (e.g., missing

at random versus missing not at random), in particular for data sets with salient spatiotemporal heterogeneity such as air pollutants time series (Junger and Ponce de Leon, 2015). PM$_{2.5}$ often exhibits evidently diurnal variation phases, which are primarily governed by local air pollutants emissions and regional meteorological conditions such as boundary layer height (Guo et al., 2017; Li et al., 2017; Huang et al., 2018; Liu et al., 2018; Miao et al., 2018; Yang et al., 2018, 2019b). Consequently, conventional

approaches like those listed in Table 1 may partially fail in accurately predicting missing values in hourly PM2.5 series.

In general, most currently available gap filling methods in Table 1 suffer from at least one of the following drawbacks: 1) partially fail for data sets with prominent gaps; 2) not self-consistent due to the requirement of supplementary data sets;3) computationally intensive (e.g., neural networks), and, most critically; 4)



unable to fairly predict daily peaks and/or minima due to the absence of essential prior knowledge of diurnal variability of monitoring targets. Given the significant heterogeneity of PM$_{2.5}$ concentration in space and time (Guo et al., 2017; Manning et al., 2018), ignoring the diurnal phases of PM$_{2.5}$ would result in large bias to the gap filled PM$_{2.5}$ data set.

**Table 1.** Overview of several popular gap filling methods to impute missingness in geophysical data sets.

| | Method | Principle or core technique | Reference |
|---|---|---|---|
| **Temporal** | Weibull | Weibull frequency distribution mapping | Nosal et al. (2000) |
| | EM | Expectation-Maximization | Junger and Ponce de Leon (2015) |
| | Interpolation | Linear regression, Spline, NAR, ARIMA, ARCH | Stauch and Jarvis (2006); Neteler (2010); Demirhan and Renwick (2018) |
| | Machine learning | Gradient Boosting, neural networks | Körner et al. (2018) Şahin et al. (2011) |
| | SSA | Imputation using singular spectrum analysis | Mahmoudvand and Rodrigues (2016) |
| | DS | Conditional resampling of a temporal subset | Dembélé et al. (2019) Oriani et al. (2016) |
| | TIMESAT | Savitzky–Golay filter, harmonic and asymmetric Gaussian functions | Jönsson and Eklundh (2004) |
| | Hybrid method | Fuzzy c-means with support vector regression and genetic algorithm | Aydilek and Arslan (2013) |
| **Spatial** | IDW | Interpolate using inverse distance weighting | Shareef et al. (2016) |
| | Kriging | Interpolate neighborhoods using Kriging | Rossi et al. (1994); Zhu et al. (2015); Singh et al. (2017) |
| | NSPI / GNSPI | Replace or interpolate with adjacent similar pixels | Zhu et al. (2012); Chen et al. (2011) |
| **Spatio-temporal** | EOF / DINEOF | Iteratively decompose and reconstruct spatial and temporal subsets using empirical orthogonal function | Beckers and Rixen (2003); Taylor et al. (2013); Liu and Wang (2019) |
| | Mosaicing | Merge numerical outputs with satellite observations | Konik et al. (2019) |
| | gapfill | Quantile regression fitted to spatiotemporal subsets | Gerber et al. (2018) |
| | STWR | Spatially and temporally weighted regression | Chen et al. (2017) |
| | SMIR | Learning machine created from historical spatial and temporal subsets | Chang et al. (2015) |
| | RFRE | Learning from other information using random forest | Bi et al. (2018); Chen et al. (2019) |

* SSA: Singular Spectrum Analysis; DS: Direct Sampling; IDW: Inverse Distance Weighting; NSPI: Neighborhood Similar Pixel Interpolator; GNSPI: Geo-statistical Neighborhood Similar Pixel Interpolator; EOF: Empirical Orthogonal Function; DINEOF: Data Interpolating Empirical Orthogonal Function; STWR: Spatially and Temporally Weighted Regression; SMIR: SMart Information Reconstruction; RFRE: Random Forest Regression

## 3 Gap filling method on the diurnal cycle constrained empirical orthogonal function

Given the significant heterogeneity of PM$_{2.5}$ diurnal phases impacted by local air pollutants emissions and atmospheric conditions, we propose to utilize the local diurnal cycle of PM$_{2.5}$ to constrain missing

value imputation for the filling of data gaps presented in the hourly time series of $PM_{2.5}$ concentration at each station. The goal is to better predict missing $PM_{2.5}$ values, especially for the daily peaks and/or minima, which are poorly predicted by conventional methods due to the absence of prior knowledge of

local diurnal phases of $PM_{2.5}$. A schematic diagram of the proposed DCCEOF method is illustrated in Figure 1. In general, the DCCEOF method consists of the following four primary steps with the goal of reconstructing the local diurnal cycle of $PM_{2.5}$ for the time series of each 24-h $PM_{2.5}$ with missingness from their discrete neighborhood fields.

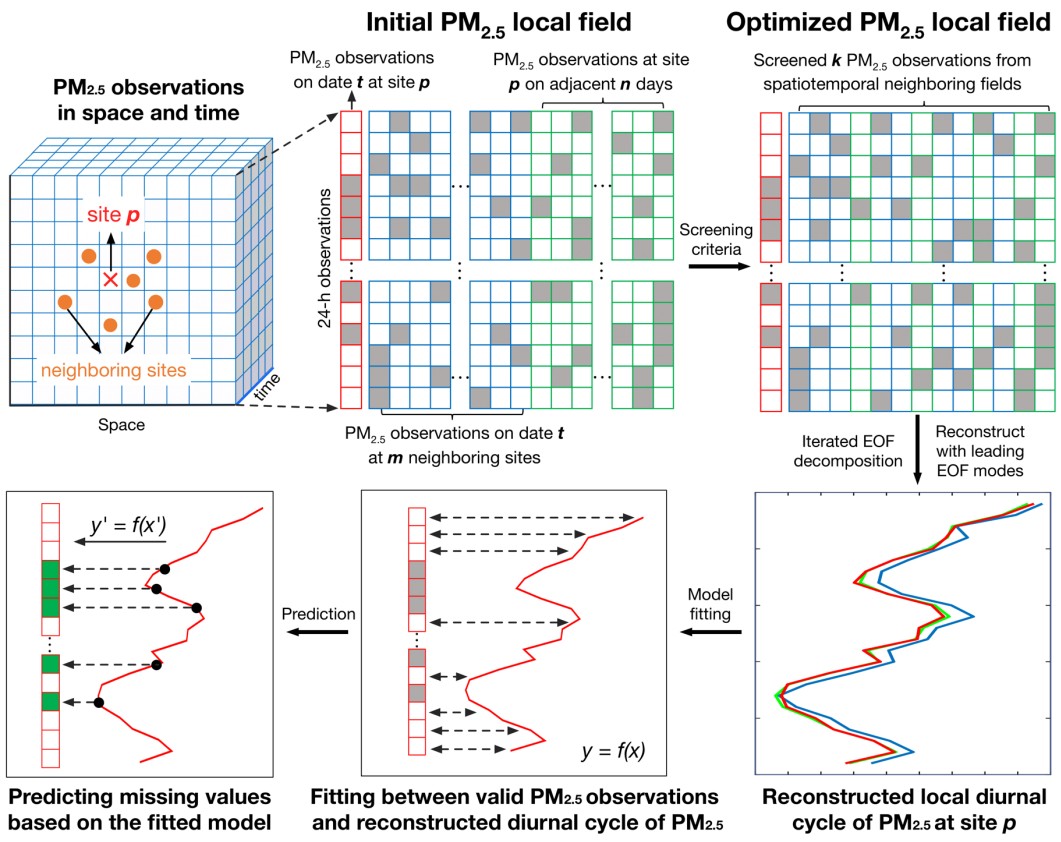

**Figure 1.** Schematic illustration of the proposed DCCEOF method for filling data gaps in hourly $PM_{2.5}$ records. The grey rectangles denote missing values.


1)  Initialize a local $PM_{2.5}$ neighborhood field: For any identified $PM_{2.5}$ missingness at site $\boldsymbol{p}$ on date $\boldsymbol{t}$ (denoted as $\boldsymbol{M_p^t}$ hereafter), an initial $PM_{2.5}$ neighborhood field in space and time (denoted as $\boldsymbol{X_{p,t}^{m,n}}$) was



first constructed using 24-h PM$_{2.5}$ observations from nearby $\boldsymbol{m}$ stations on date $\boldsymbol{t}$ and observations from adjacent $2\boldsymbol{n}$ days at site $\boldsymbol{p}$. Mathematically, the neighborhood field $X_{p,t}^{m,n}$ can be expressed as:

$$X_{p,t}^{m,n} = \{x_t^1, x_t^2, \dots, x_t^m; \ x_p^{t-n}, \dots, x_p^{t-2}, x_p^{t-1}, x_p^{t+1}, x_p^{t+2}, \dots, x_p^{t+n}\} \tag{1}$$

It is clear that $\boldsymbol{m}$ and $\boldsymbol{n}$ are two critical factors in determining the dimension of $X_{p,t}^{m,n}$ as a smaller $\boldsymbol{m}$ and $\boldsymbol{n}$ would yield a more compact and localized PM$_{2.5}$ neighborhood field. Considering a too compact neighborhood field may be insufficient to reconstruct the local diurnal cycle of PM$_{2.5}$ fairly due to limited information, since missingness may also present in each candidate 24–h PM$_{2.5}$ concentration time series. $\boldsymbol{m}$ was defined as the number of stations within 100 km of the target station and $\boldsymbol{n}$ was set to 7 (i.e., one week before and after date $\boldsymbol{t}$ respectively) in our algorithm. This configuration resulted in adequate samples for the construction of $X_{p,t}^{m,n}$ while rendering the computational workload manageable.

2) Construct a compact PM$_{2.5}$ neighborhood field: Since the initial PM$_{2.5}$ neighborhood field $X_{p,t}^{m,n}$ might include many irrelevant observations with different diurnal phases given large spatial and temporal intervals (i.e., $\boldsymbol{m}$ and $\boldsymbol{n}$), a compact neighborhood field should be constructed by only retaining observations that are highly related to the target PM$_{2.5}$ time series $x_p^t$ most critically, with similar diurnal phases. Therefore, the covariance rather than correlation between the target time series $x_p^t$ and every candidate PM$_{2.5}$ time series in $X_{p,t}^{m,n}$ was first calculated (normalized by the number of valid data pairs, i.e., without missingness). Subsequently, the candidate PM$_{2.5}$ time series were sorted with respect to the magnitudes of covariances in a descending order. Finally, the first $\boldsymbol{k}$ time series were retained to construct the optimized PM$_{2.5}$ neighborhood field $\widehat{X^k}$ by complying with the criterion that there are at least five valid observations at each specific time (i.e., observations in each row) from 00:00 to 23:00. The aim of this configuration is to avoid large bias in the subsequent diurnal cycle reconstruction using EOF, since large outliers may emerge at times without any valid observation. Mathematically, the process to construct $\widehat{X^k}$ can be formulated as follows:

$$C_{x'} = COV(x_p^t, x' | X_{p,t}^{m,n}) \tag{2}$$

$$\widehat{X^k} = \{x_1', x_2', \dots, x_k' \ | C_{x_k'} < C_{x_{k-1}'} < \cdots < C_{x_1'}\} \tag{3}$$

where $x'$ denotes the time series of candidate PM$_{2.5}$ in $X_{p,t}^{m,n}$ and $COV$ is the covariance function.



3)  Reconstruct the spatiotemporally localized diurnal cycle of PM$_{2.5}$: The diurnal cycle of PM$_{2.5}$ at site $\boldsymbol{p}$ on date $\boldsymbol{t}$ (denoted as $\boldsymbol{\beta_p^t}$) was then reconstructed from the optimized neighborhood field $\widehat{X^k}$ using EOF in an iterative process similar to the DINEOF method (Beckers and Rixen, 2003). In our DCCEOF method, the time series of the target PM$_{2.5}$ $\boldsymbol{x_p^t}$ were also included as a basic constraint for the reconstruction of the local diurnal cycle of PM$_{2.5}$ $\boldsymbol{\beta_p^t}$ and the whole field was then denoted as $\widetilde{X}$.

$$\tilde{X} = \{x_p^t, \widehat{X^k}\} \tag{4}$$

The EOF-based gap filling process can be outlined as follows: a) 20% of valid PM$_{2.5}$ observations in $\widetilde{X}$ were first retained for cross validation (CV) and then data values at these points were treated as gaps by replacing with nulls (i.e., missing value); b) given that a small amount of missing values would not significantly influence the leading EOF mode for the original data set, we may assign a first guess (here we used the mean value of valid data in each column) to the data points where missing values are identified to initialize the EOF analysis; c) EOF analysis was performed on the previously generated matrix (i.e., gaps are filled with column mean) in a form of singular value decomposition (SVD) and then data values at value-missing points were replaced by the reconstructed values at the same points using the first EOF mode. These processes can be expressed as:

$$[U, S, V] = svd(< \tilde{X} >) \tag{5}$$
$$X' = u_1 * s_1 * v_1 \tag{6}$$

where $< \tilde{X} >$ denotes the initial matrix in which the missing values were filled with column means. U, S, and V are three matrices derived from SVD while $u_1$, $s_1$, and $v_1$ denote the SVD components in the first EOF mode. $X'$ is the reconstructed matrix using the first EOF mode; e) iteratively decompose and reconstruct the matrix while updating data values at the value-missing points using the first EOF mode till the convergence is confirmed by the mean square error at each iteration; f) repeat the previous iterative processes for the following EOF modes till the final convergence (i.e., error starts to increase as the new EOF mode is included). The $\boldsymbol{\beta_p^t}$ was finally obtained by standardizing the identified leading EOF modes.

4)  Missing value imputation: Finally, a linear relationship was established between valid PM$_{2.5}$ observations in $\boldsymbol{x_p^t}$ and the corresponding values in $\boldsymbol{\beta_p^t}$. Missing values in the time series of the original





$PM_{2.5}$ were then predicted by mapping data values in the reconstructed diurnal cycle at missing time based on the established linear relationship.

In general, the proposed DCCEOF gap filling method is a univariate and self-consistent method since no additional data record is required for missing value imputation. Rather, the method works by relying primarily on the local diurnal cycle of $PM_{2.5}$ that can be reconstructed from discrete $PM_{2.5}$ neighborhood fields in space and time. Compared with conventional gap filling methods that work on a statistical basis (e.g., spline interpolation), the unique feature and novelty of the proposed DCCEOF method lies in its utilization of the diurnal cycle to constrain the missing value imputation, rendering physically meaningful predicted values with high accuracy.

## 4 Demonstrative case study in China

### 4.1 China in-situ $PM_{2.5}$ concentration records

The near surface mass concentrations of $PM_{2.5}$ across China are measured primarily using the tapered element oscillating microbalance analyzer and/or the beta-attenuation monitor at each monitoring station. The instruments' calibration, operation, maintenance, and quality control are all properly conducted by complying with the China Environmental Protection Standards of GB3095-2012 and HJ 618–2011. $PM_{2.5}$ concentrations are measured by these instruments with an accuracy of $\pm 5$ μg/m$^3$ for ten-minute averages and $\pm 1.5$ μg/m$^3$ for hourly averages (Guo et al., 2017; Miao et al., 2018). Although the hourly $PM_{2.5}$ observations in China have been publicly available since 2013, the $PM_{2.5}$ records used in the present study were retrieved following May 2014 via a web crawler program.

Figure 2 depicts the spatial distribution of the national ambient air quality monitoring network in China as well as the start year for the first release of $PM_{2.5}$ measurements at each individual station. Given the fact that our data were retrieved following May 2014, stations deployed before 2014 are hard separate from those being built in 2014 and hence, they were all designated the same way in Figure 2. At present, this network consists of more than 1,600 stations, in which about 940 stations were established before 2015. The total number of stations was increased to 1,494 in June 2015, and then only four stations were




newly deployed in the following one and half years until December 2016. In other words, the vast majority (92.4%) of PM$_{2.5}$ stations in the current monitoring network were established before the middle of 2015.

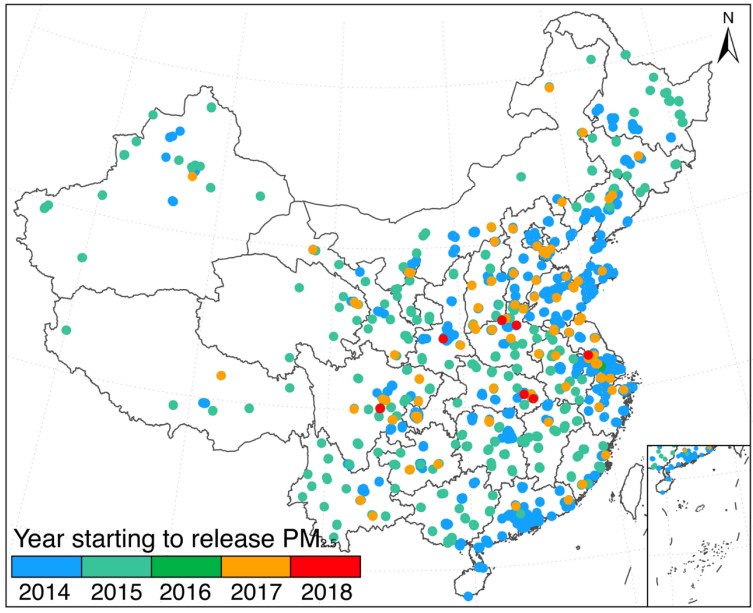

**Figure 2.** Spatial distribution of China's national ambient air quality monitoring stations from May 2014 to April 2019. Circles with distinct color indicate the year in which the first PM$_{2.5}$ observation was publicly released at each station in our used data record.

## 4.2 Results and discussion

### 4.2.1 Data completeness of in-situ PM$_{2.5}$ records across China

The features of data gaps presented in the retrieved hourly PM$_{2.5}$ concentrations were first evaluated. Figures 3a–c present the daily averaged missing value ratio, the occurrence frequency of missingness (defined as the ratio of days with missing values presented in 24-hour PM$_{2.5}$ observations (regardless of the number of missing values) divided by the total number of days since the release of the first PM$_{2.5}$ observation), and the diurnal phases of the most frequently occurring missing values at each monitoring station since the first release of PM$_{2.5}$ observations to the public, whereas Figures 3d–f show the corresponding histograms, respectively. Note that most of stations exist daily-averaged missing value

ratios less than 10% (Figure 3a). Nonetheless, prominent data gaps are still observed at several monitoring stations (red dots in Figure 3a) with more than 70% of hourly PM$_{2.5}$ observations lost in daily 24-h

measurements. After checking the retrieved PM$_{2.5}$ data records across these stations, we found that most of these stations stopped releasing PM$_{2.5}$ observations after the middle of 2015.

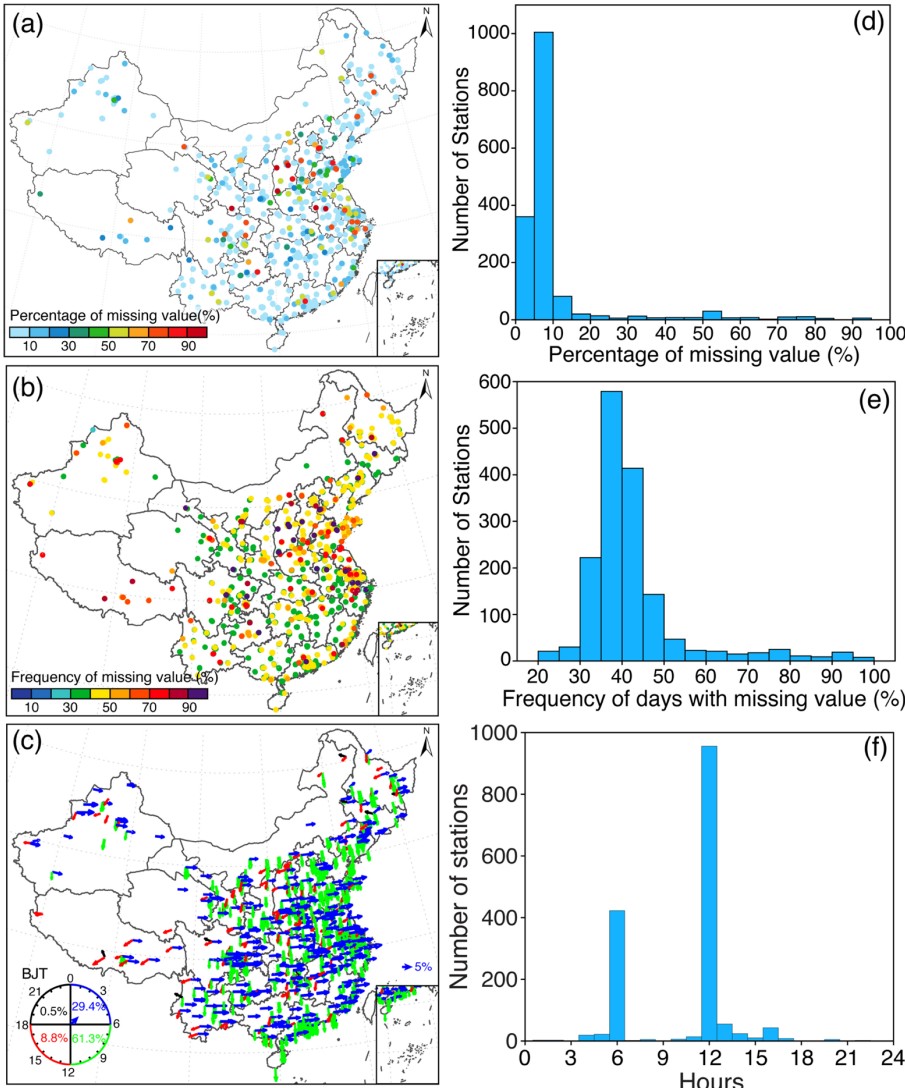

**Figure 3.** Frequency of missing values present in hourly PM$_{2.5}$ records at each station since the first release of PM$_{2.5}$ observations onward. (a) Frequency of days with missing values, (b) diurnal phases of

maximum occurring frequency of missing values, (c) and (d) are corresponding histograms for (a) and (b), respectively. The arrow direction denotes the local time (Beijing time, BJT) at which missing values





occurred most frequently and the arrow length indicates the magnitude of frequency. The varying diurnal phases of missing values were represented by different colors: blue (00~06 BJT), green (06~12 BJT), red (12~18 BJT), and black (18~24 BJT).


Despite the small magnitudes (~10%) of daily-averaged missing value ratios (Figure 3d), data gaps in our retrieved hourly PM$_{2.5}$ records are still salient, which is evidenced by the occurrence frequency of missing values in daily PM$_{2.5}$ observations (Figure 3b). In contrast to the daily averaged missing value ratios (Figure 3a), the missing value frequency has a relatively larger magnitude (~40%), revealing that data

gaps occurred frequently in the retrieved PM$_{2.5}$ records, as four out of ten days PM$_{2.5}$ samplings were subject to data gaps (Figure 3e). Therefore, there is an urgent need to fill in the data gaps in China PM$_{2.5}$ records to facilitate the exploitation of these valuable records.

In addition, the diurnal phases of the occurrence of missing values were examined. Figure 3c presents the detailed time (represented by the arrow direction) and frequency (represented by the relative length of

each arrow) of the most frequently occurring missing values, whereas Figure 3f shows the histogram of the local time at which missing values occurred most frequently at each monitoring station. It can be found that the missing values occurred more frequently in the morning for most stations (90.7% of total population of stations), particularly at 0600 and 1200 of the Beijing time, while the possible reason for which remains unclear.

**4.2.2 Impacts of missing values on daily-averaged PM$_{2.5}$**

It is well known that the number of missingness is highly linked to how well the estimated PM$_{2.5}$ daily averages and their associations with application results can be trusted. As such, the possible impacts of PM$_{2.5}$ missing values were examined to provide a holistic viewpoint of the adverse impacts of data gaps, given the fact that the daily averages are frequently used in many PM$_{2.5}$-related studies. First, gap–free

observations of hourly PM$_{2.5}$ within 24h were extracted. Since sampling based on all enumerated combinations for the given number of missing values is undoubtedly time consuming, we randomly sampled 1,000 days from all gap-free days observations, especially for different pollution scenarios (clean versus polluted, respectively) in order to make the workload manageable. In addition, days with daily-



averaged $PM_{2.5}$ lower than the 10th quantile of all gap-free days were considered as clean scenario, while
those greater than the 90th quantile were treated as polluted scenario. Subsequently, a varying number
(range from 1 to 23) of data values were treated as gaps in every daily $PM_{2.5}$ observation randomly and
then mean relative differences (MRDs) in daily-averaged $PM_{2.5}$ derived from between hourly records with
and without data gaps were calculated as a measure to evaluate the potential impacts of missingness.

Figure 4a shows the estimated MRDs at the 10th, 50th, and 90th quantiles for different numbers of missing
values in 1,000 randomly sampled 24–h $PM_{2.5}$ observations, indicating that larger biases could be
introduced to the daily averages with the increase in the total missingness. Given the symmetrical
behavior of MRDs around zero (like a Gaussian distribution) for each given number of missingness, we
may infer that random biases could be introduced into $PM_{2.5}$ daily averages if missing values are ignored
for the calculation of daily averages of $PM_{2.5}$. These random biases, in turn, could yield large uncertainties
to the subsequent results such as trend estimations. To further evaluate the impacts of missingness on
daily averages of $PM_{2.5}$, in particular at different pollution scenarios, MRDs were calculated on 1,000
clean and polluted days, respectively (Figure 4b–d). On average, MRDs vary with larger deviations for a
given number of missingness on clean days than on polluted days (Figure 4b). Regarding MRDs at 10th
and 90th quantiles, we may deduce that missing values would result in larger bias to $PM_{2.5}$ daily averages
on clean days than in polluted conditions given larger MRDs for clean scenarios (Figures 4c–d). This
effect is in line with expectations since $PM_{2.5}$ concentrations often exhibit larger diurnal variations on
cleaner days and smaller deviations on polluted days due to the boundary layer height (BLH) effect (Li
et al., 2017; Miao et al., 2018). Note that six missing values would result in as large as approximately 5%
of bias (10% for 12 missing values) to daily averages of $PM_{2.5}$ during clean days (Figures 4c-d).

In addition to the number of missing values, possible impacts of diurnal phases of missing values on
daily-averaged $PM_{2.5}$ were also examined (Figure 5). Different diurnal phases were observed for MRDs
associated with missingness at different pollution levels. Missing values in the afternoon and evening
would more likely result in overestimations to daily-averaged $PM_{2.5}$, whereas underestimations for
missingness in the morning and night. Moreover, the missingness in the afternoon during clean days has
a larger potential to overestimate daily-averaged $PM_{2.5}$ than at other times. This effect could be largely
associated with the diurnal phases of $PM_{2.5}$ as daily peaks are oftentimes observed in the early morning



(Wang and Christopher, 2003), though such a diurnal variation pattern may differ by regions (Lennartson et al., 2018). Furthermore, the diurnal phases of $PM_{2.5}$ are largely dominated by the diurnal variation of regional emissions and boundary layer processes (Guo et al., 2016; Lennartson et al., 2018; Miao et al., 2018;Yang et al., 2019b). In contrast, the diurnal phases of MRDs are not evident during polluted days. All these findings collectively suggest the need to fill in data gaps presented in hourly $PM_{2.5}$ observations, especially for those measured during clean days, since missing values would result in larger biases to daily-averaged $PM_{2.5}$ than those during polluted phases.

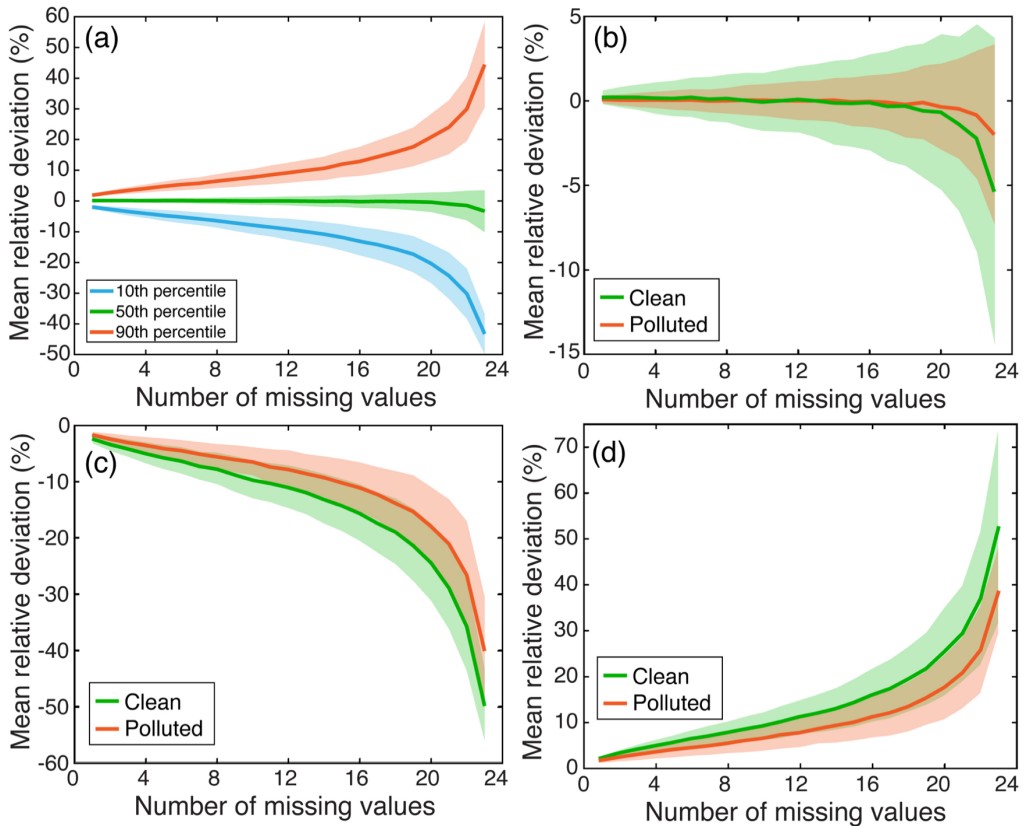

**Figure 4.** Impacts of the number of missing values on daily averages of $PM_{2.5}$. Mean relative deviations were calculated between $PM_{2.5}$ daily averages estimated from hourly records with a given number of missing values and the original one without missing values. (a) Deviations at different percentiles at all-sky conditions; (b) deviations at the 50th percentile under different pollution scenarios; (c) same as (b) but for the 10th percentile; (d) same as (b) but for the 90th percentile. Thick lines represent mean deviations while shaded regions are uncertainties of one standard deviation from the mean at each side.



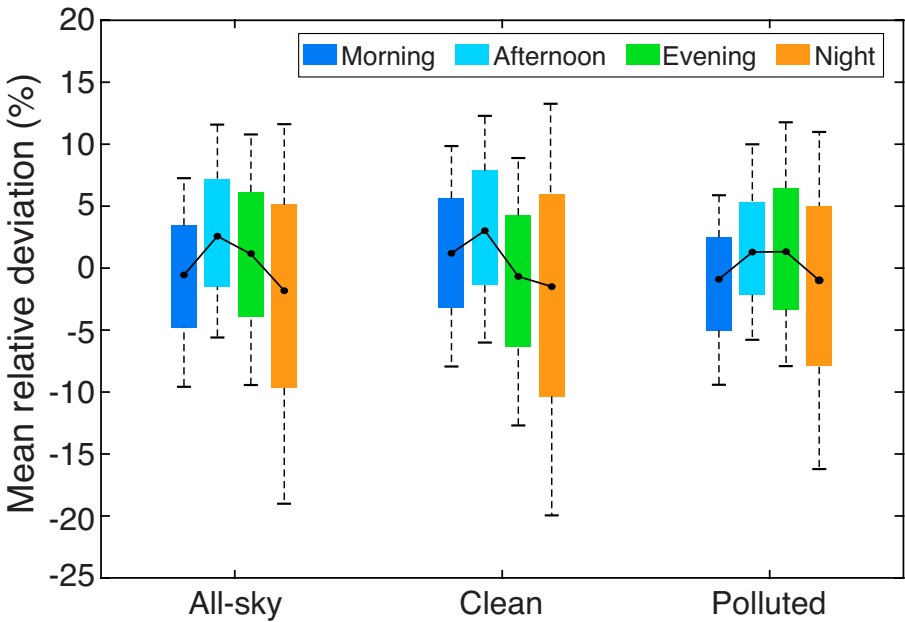

**Figure 5.** Impacts of diurnal phases of missing values on PM$_{2.5}$ daily averages. Hourly PM$_{2.5}$ values in the morning (07~11 BJT), afternoon (12~16 BJT), evening (17~21 BJT), and night (22~06 BJT) were removed from the original hourly PM$_{2.5}$ time series throughout the day to resemble missing values respectively. On each box, the black dots represent medians of mean relative deviations while the bottom and top edges of the box indicate the 25th and 75th percentiles and the whiskers extend to the 10th and 90th percentiles, respectively.

### 4.2.3 Performance of DCCEOF method

Since the goal of the proposed DCCEOF method is to reconstruct the diurnal cycle of PM$_{2.5}$ from a spatiotemporally localized neighborhood field even in the presence of data gaps, three gap-free 24-h PM$_{2.5}$ observations at different pollution levels were selected at two different monitoring stations (with different numbers of neighboring stations within 100 km of the target station) respectively to assess the efficacy of the proposed DCCEOF gap filling method. As shown in Figure 6, the DCCEOF method performed well in reconstructing the local PM$_{2.5}$ diurnal cycles from the discrete neighborhood field, and the reconstructed diurnal variation patterns were highly in line with the practical observations. In particular, the DCCEOF method enabled us to successfully restore the missing PM$_{2.5}$ information even at the

inflection times, e.g., the peak value in Figure 6c and the minimum value in Figure 6e, which are oftentimes hard to recover by statistical interpolation approaches. Nonetheless, compared with practical

PM$_{2.5}$ observations, the reconstructed PM$_{2.5}$ diurnal cycle was still unable to sufficiently restore all types of local variations (e.g., PM$_{2.5}$ observations between 0700 and 1100 shown in Figure 6f). This is consistent with our initial understanding because PM$_{2.5}$ concentrations vary significantly in space and time. Moreover, the reconstructed PM$_{2.5}$ diurnal cycle is derived from a limited number of leading EOF modes and hence it only captures the dominant variation patterns of the neighborhood field while some local

variations are ignored. In spite of this potential drawback, the proposed DCCEOF method still exhibited high accuracy in restoring the local PM$_{2.5}$ diurnal cycle from a discrete neighborhood field.

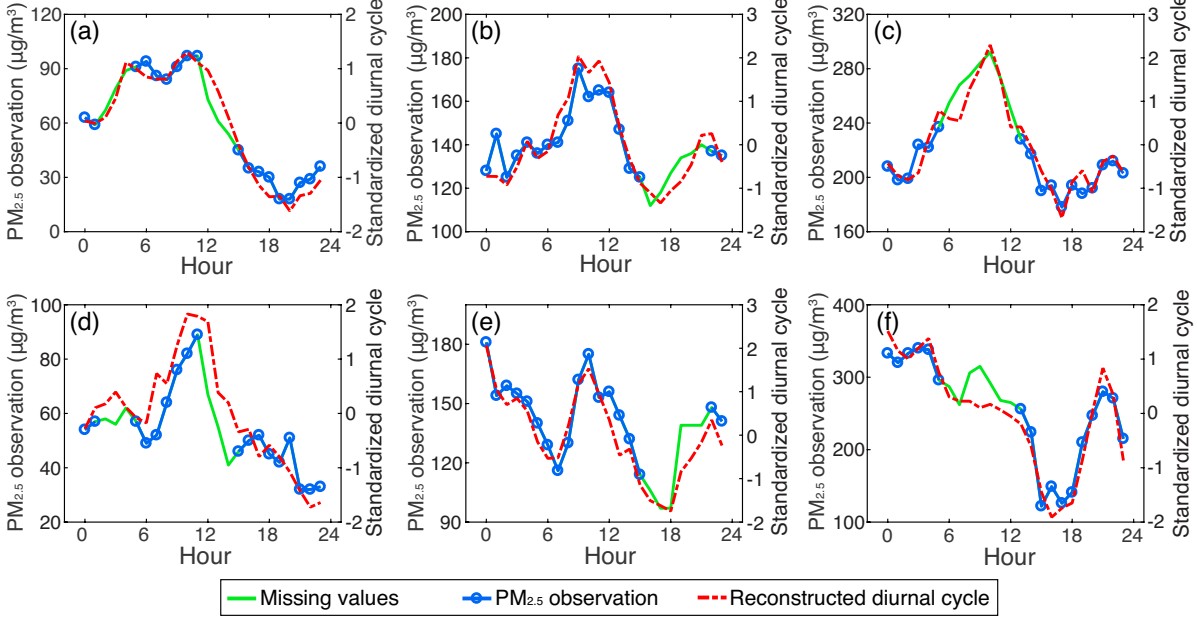

**Figure 6.** Comparisons of practical PM$_{2.5}$ concentrations with the reconstructed spatiotemporally localized PM$_{2.5}$ diurnal cycles at different pollution levels. For each trial, 6 valid PM$_{2.5}$ observations were

treated as missing values to simulate gapped PM$_{2.5}$ time series prior to diurnal cycle reconstruction for a given day. Note the number of neighboring stations differs between these two cases (58 for the top panel and 16 for the bottom).





To assess the performance of the proposed DCCEOF gap filling method, we retrieved the hourly PM$_{2.5}$ observations recorded at one monitoring station in Beijing during the time from August 1 to 7, 2014 and then some valid observations were treated as missing values for the subsequent gap filling. The DCCEOF method performed better than the conventional spline interpolation approach in restoring the artificially masked missing values, especially for those at the inflection times at which spline interpolation failed to predict with good accuracy (Figure 7). However, both methods failed in predicting the minimum values on August 2. After manually checking the original data records, we found that the local variation of PM$_{2.5}$ at this station differed largely from that of neighboring stations at the same time.

Figure 8 presents a more general evaluation of the prediction accuracy of the proposed DCCEOF gap filling method, which compares the predicted values with the retained data values at different pollution levels. It indicates that the proposed method has good imputation accuracy, with a CV correlation coefficient of 0.82 on clean days (Figure 8a) and 0.95 for polluted days (Figure 8b). As stated earlier, higher imputation accuracy is expected for filling gaps on polluted days than cleaner days given the less dynamic features of PM$_{2.5}$ concentrations on polluted days. This is also evidenced by the scatter plot shown in Figure 8a, in which larger variance is observed between the predicted values and the practical PM$_{2.5}$ observations. This effect also reveals the larger spatiotemporal heterogeneity of PM$_{2.5}$ concentrations in clean scenarios.

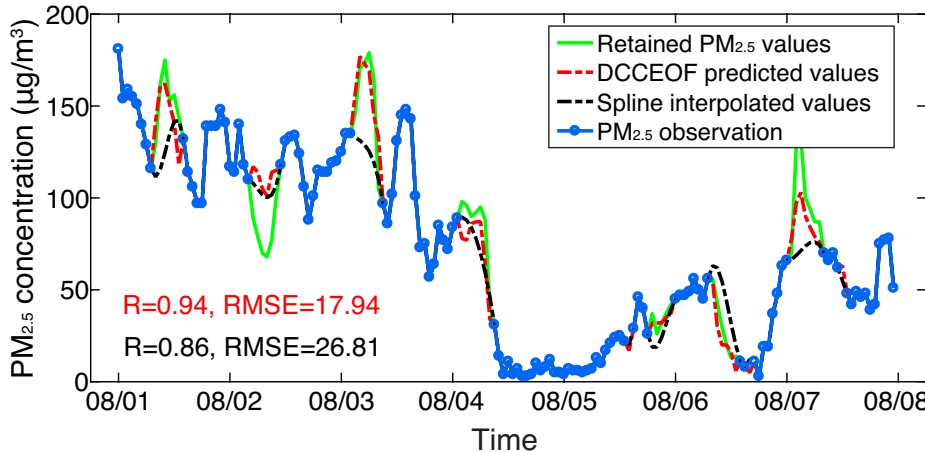

**Figure 7.** Comparison of gap filled hourly PM$_{2.5}$ time series reconstructed using spline interpolation and the proposed diurnal cycle prescribed gap filling method at the Wanshou Temple station in Beijing





between 1 and 7 August 2014. The green line shows the practical PM$_{2.5}$ observations that were treated as gaps while their original values were retained for cross validation.

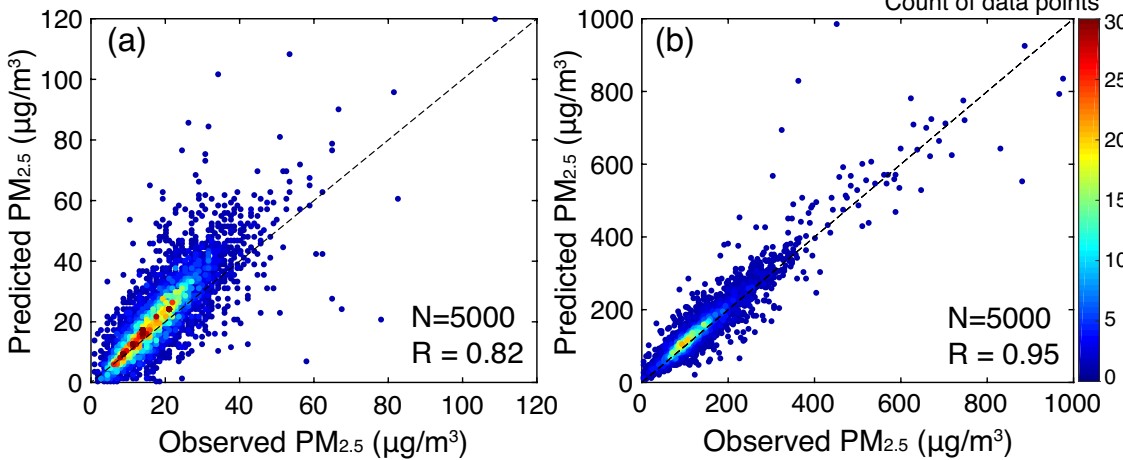

**Figure 8.** Comparisons of PM$_{2.5}$ observations with the reconstructed data values during clean (a) and polluted (b) phases. For each scenario, the results were derived from 1,000 days of gap-free PM$_{2.5}$ observations with 5 valid values which were randomly retained from 24-h observations on each sampled date for cross validation.

Given the DCCEOF method can work well by relying primarily on the spatiotemporally localized neighborhood field to reconstruct the local PM$_{2.5}$ diurnal cycle for the subsequent missing value imputation. Note that the number of missing values and the population of neighboring stations are two critical factors to fill gaps via the DCCEOF method. Therefore, sensitivity experiments were performed

to quantify the response of prediction accuracy to the variation of these two parameters. Figure 9a shows the response of prediction accuracy (in terms of correlation coefficient) of the proposed method to the varying number of missing values in each sampled time series of 24–h PM$_{2.5}$. It is clear that the prediction accuracy generally decreases with the increase of the number of missing values. This effect can be attributable to the fact that the target PM$_{2.5}$ time series is applied as a critical constraint for the screening

of candidate PM$_{2.5}$ observations in space and time to construct the spatiotemporally localized neighborhood field for the reconstruction of the local PM$_{2.5}$ diurnal cycle. Consequently, more missingness would make the constructed neighborhood field have large uncertainties due to less





information being left for the selection of related time series of PM$_{2.5}$, which in turn undermines the overall accuracy of the predicted results.

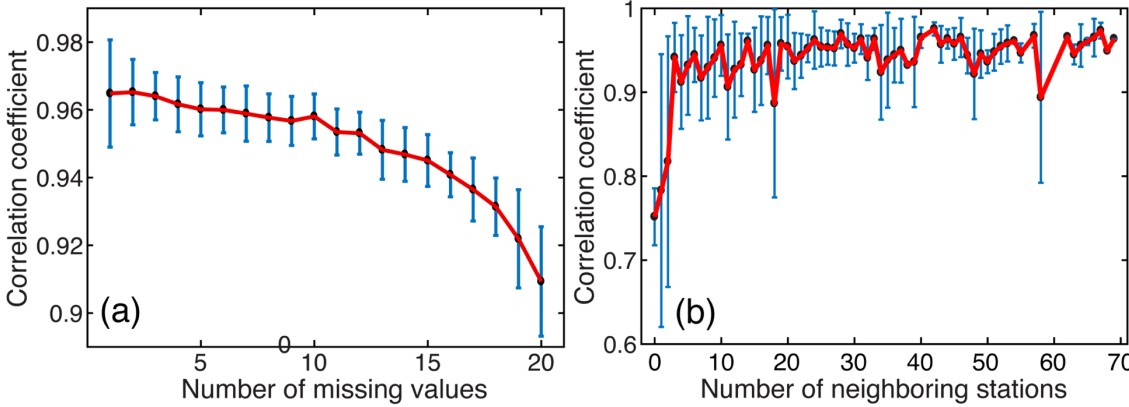

**Figure 9.** Impacts of the number of missing values present in hourly PM$_{2.5}$ records for every 24-h (upper panel) and the total number of neighboring stations (bottom panel) on the performance of the proposed gap filling method. The error bars denote one standard deviation of each value from the mean on each side.

Figure 9b presents the potential impacts of the total number of neighboring stations on the prediction accuracy at the target station. The total number of neighboring stations within 100 km of the target station was first calculated and then sensitivity experiments were performed for each selected number of neighboring stations. Specifically, ten stations were randomly selected for each given number of neighboring stations within 100 km, and then 20 gap-free PM$_{2.5}$ observations were sampled at each individual station. For each gap-free PM$_{2.5}$ observation within 24-h, six values were retained and then treated as gaps for cross validation.

It is indicative that the DCCEOF method would yield high prediction accuracy with an adequate number of neighboring stations, as three neighboring stations would render promising prediction accuracy (Figure 9b). Large biases would be introduced with a limited number of neighboring stations (<3) due to the lack of sufficient prior information for the reconstruction of the local PM$_{2.5}$ diurnal cycle. In general, the prediction accuracy may be improved with the increase of the number of neighboring stations but the enhancement effect is not obvious at those stations with more than three neighboring stations.



Nonetheless, the present results indicate that the increase of neighboring stations would reduce the
uncertainties in the final predicted values, as evidenced by smaller standard deviations of correlation coefficients for cases with more neighboring stations. Moreover, diurnal cycle reconstructed from the neighborhood field in space is more accurate than using $PM_{2.5}$ observations from adjacent times, which is evidenced by smaller correlation values with limited neighboring stations.

Figure 10 presents the benefits of the DCCEOF method for in–situ hourly $PM_{2.5}$ records at each individual
monitoring station in terms of the improvement of the data completeness ratio as well as the reduction of gap frequency. It shows that the DCCEOF method enables the improvement of the data completeness ratio of hourly $PM_{2.5}$ records by about 5% on average at the national scale, and the overall data completeness ratio has been improved from 89.2% to 94.3% (Figure 10a). Despite the small magnitude of the data completeness improvement ratio, the occurrence frequency of days with missingness has been
prominently reduced, with the averaged frequency of days with missingness declined from 42.6% to 5.7% (Figure 10b). In general, the gap-filled $PM_{2.5}$ record via the DCCEOF method is more temporally complete and thus can be used as a good data source for further $PM_{2.5}$-related studies.

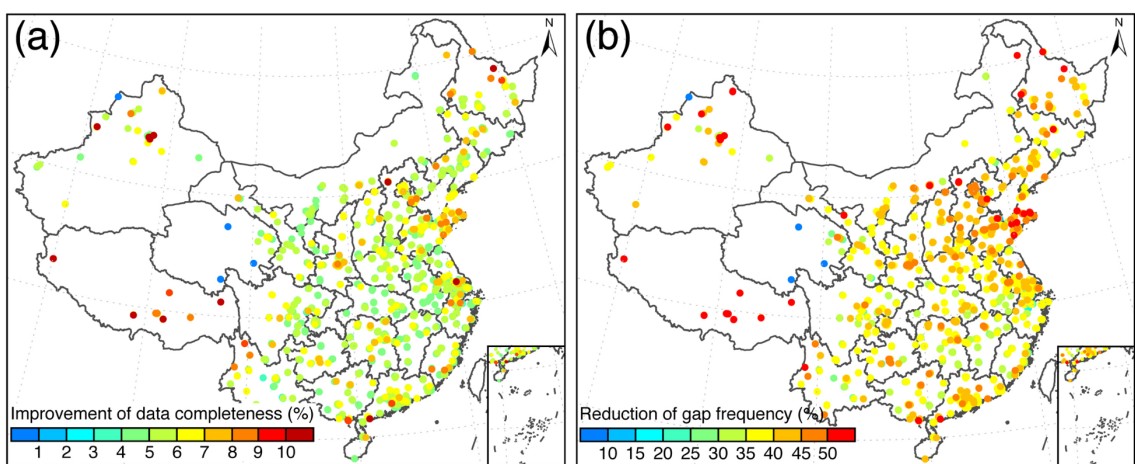

**Figure 10.** Benefits of the proposed gap filling method applied to China in-situ hourly $PM_{2.5}$ records at
each individual monitoring station. (a) Improvement of data completeness and (b) reduction of the percentage of days with missing values.



## 5. Conclusions

A practical and realistic gap filling method termed DCCEOF is proposed in the present study to cope with missingness in time series with significant diurnal variability. Compared with the conventional gap filling
methods, the proposed DCCEOF method is self–consistent, physically meaningful, and more accurate, given the utilization of the reconstructed spatiotemporally localized diurnal cycle to constrain the missing value imputation. Such an endeavor enables the proposed gap filling method to predict missing values even at inflection times, like daily peaks or minima, with good accuracy.

As a demonstration, the proposed DCCEOF method was practically applied to fill in data gaps in hourly
$PM_{2.5}$ data records that were acquired from China's national air quality monitoring network, and the cross-validation results indicate a promising prediction accuracy of the proposed DCCEOF gap filling method in restoring $PM_{2.5}$ missingness. The method performs even better in predicting missing values during polluted phases rather than during clean days given smaller variations of $PM_{2.5}$ concentrations in space and time. Further sensitivity experiments suggest that the overall accuracy of the DCCEOF method would
slightly decrease with the increase of the amount of missingness in daily 24-h $PM_{2.5}$ observations. This effect is largely associated with larger uncertainties in the construction of spatiotemporally localized $PM_{2.5}$ neighborhood fields. In addition, an adequate number of neighboring stations in space is essential to the final prediction accuracy of missing value imputation. The experimental results suggest that three neighboring stations within 100 km to the target station would yield a promising prediction accuracy, and
the more neighboring stations, the less the uncertainties of the predicted values.

Moreover, the data gaps presented in our retrieved in–situ hourly $PM_{2.5}$ records were explored. In general, the missingness ratio is less than 10% at most stations across China. Meanwhile, data gaps occur more frequently at 0600 and 1200 BJT than other time. After gap filling, the data completeness ratio of China in–situ hourly $PM_{2.5}$ record was improved to 94.3% while the frequency of days with missingness was
markedly reduced from 42.6% to 5.7%. The gap filled hourly $PM_{2.5}$ record can thus be used as a promising data source for better $PM_{2.5}$ concentration mapping at the national scale, e.g., incorporating in-situ $PM_{2.5}$ information from neighboring stations to advance $PM_{2.5}$ prediction accuracy.

Overall, the proposed DCCEOF gap filling method provides a realistic and promising way to deal with missingness presented in hourly $PM_{2.5}$ concentration records which oftentimes exhibit pronounced diurnal



phases. Given its self-consistent nature, this method can be thereby directly applied to PM$_{2.5}$ datasets measured in other regions and/or other time series of other data with similar barriers. A more general comparison of this method with many other conventional gap filling methods will be conducted in the future to further evaluate the performance and accuracy of the DCCEOF method in handling various types of data gaps.

**Acknowledgments**

The authors acknowledge the China National Environment Monitoring Centre (http://www.cnemc.cn) for making the essential hourly PM$_{2.5}$ concentration measurements publicly available. This study was supported by the National Natural Science Foundation of China under grants 41701413, 41771399 and 91544217, and the Shanghai Sailing Program under grant 17YF1404100.

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
