# Peer review of "Filling the gaps of in-situ hourly PM2.5 concentration data with the aid"

_Atmospheric Measurement Techniques, 2019_

## Referee Comment (RC1) · Anonymous Referee #1 · 30 Sep 2019

Comments on "Filling the gaps of in-situ hourly PM$_{2.5}$ concentration data with the aid of empirical orthogonal function constrained by diurnal cycles"

The method recommended by the authors for the missing value filling to hourly PM$_{2.5}$ data is interesting. It could be useful for relevant study.

Some concerns remain as following, which might be considered to further improve the method.

(a) Because the PM$_{2.5}$ diurnal variation could vary largely from day to day, is it possible that some typical classification of PM$_{2.5}$ diurnal variation could be established and considered, which should be helpful if one can determine the general pattern of PM$_{2.5}$ diurnal variation for the interested day and make more adequate filling for the missing PM$_{2.5}$ data.

(b) The PM$_{2.5}$ diurnal variation could be related to some specific meteorological factors as well as their diurnal evolution. Is it possible that the diurnal variation of specific meteorological factors be considered within the authors recommended missing value filling method?

(c) What is the applicability of the method? Especially for the different spatial distribution of the air quality monitoring stations which are condense over eastern China but sparse over western part of the country.

(d) In the manuscript, the authors made cross validation for missing value filling for several hours, is it possible that there are missing value for a specific station for one day or several days? If this situation happens, how about the performance of the authors recommended method to make missing value filling?

Some specific comments are also listed below for the authors.

1. Line 60, "data cleaning processes", consider using more accurate wording to describe what the authors want to mention.

2. Lines 70-71, it is better to directly give the disadvantages of "approaches of ignoring missing values or excluding records on days with missing values", rather than arbitrarily comment these approaches as "unreasonable".

3. Table 1, the lines for the references are not quite clear, it is difficult to find which reference is corresponding to which method.

4. Line 152, "$m$ was defined as the number of stations within 100 km of the target station", as the authors mentioned about the "significant heterogeneity" of the $PM_{2.5}$ data, is the setting of "100 km" improperly greater in this context? $PM_{2.5}$ concentration can vary largely even within a small area.

   Moreover, the air quality monitoring stations are densely distributed over eastern China but sparsely over western part of China. Is there any special consideration should be taken on this issue?

5. The day-to-day $PM_{2.5}$ diurnal variation could vary largely, which depends on whether it is a clean day or a severe polluted day, as well as the various weather conditions. The authors also mentioned this in Lines 302-304. While the method the authors suggested only considers the diurnal variation of one week before and one week after the data missing day to be filled. Is it possible any variety in the diurnal variation of $PM_{2.5}$ can be considered in the recommended method? Also, more detailed classification and establishment of the typical patterns of $PM_{2.5}$ diurnal variation and adequate consideration of this issue could be very helpful to improve the data filling method suggested.

6. Figure 3, it is a little difficult to understand the variables illustrated. The result presented in each panel of the figure seems not match with the caption. The name of the x axis in Figure 3f could be better as "hour".

7. Figure 4a, the 50th percentile of the mean relative differences generally remains constant around zero, does this mean that the 50th percentile is subjective of less influence from missing values?

8. Figure 6, the reconstructed diurnal $PM_{2.5}$ variation seems to be a smoothed average of the observations near the interested station within a week before and after the interested day, it cannot reconstruct any particular variation of $PM_{2.5}$ such as those at 19:00 local time in Figure 6e and at 08:00-09:00 local time in Figure 6f.

9. Lines 409-411, because of the "significant heterogeneity" of the $PM_{2.5}$ spatial distribution, how about the spatial distribution of the diurnal pattern of $PM_{2.5}$

variation? Is it practical to consider the variability of $PM_{2.5}$ at the stations 100 km away to fill missing value of $PM_{2.5}$?

10. Do Figure 10a and 10b reflect the same information from different perspectives? Is it possible just keep one figure to discuss the issue?

11. Lines 414-422 and Figure 10, have the authors done data filling for all the available $PM_{2.5}$ data over China with the recommended method? Is the evaluation presented here are based on data filling for the whole dataset of $PM_{2.5}$ available?

---

## Referee Comment (RC2) · Anonymous Referee #2 · 7 Nov 2019

[referee-annotated manuscript omitted]

---

## Author Response (AR1)

**Response to referees' comments:**

**Referee #1**

The method recommended by the authors for the missing value filling to hourly  $PM_{2.5}$  data is interesting. It could be useful for relevant study.

**Reply:** Thank you for your insightful comments and valuable suggestions which help a lot in improving the manuscript. All your raised concerns (in black) have been properly and adequately addressed in our revised manuscript and point-to-point responses (in blue) can be found below.

Some concerns remain as following, which might be considered to further improve the method.

(a) Because the PM2.5 diurnal variation could vary largely from day to day, is it possible that some typical classification of PM2.5 diurnal variation could be established and considered, which should be helpful if one can determine the general pattern of PM2.5 diurnal variation for the interested day and make more adequate filling for the missing PM2.5 data.

**Reply:** Thanks for your constructive suggestions. Actually, what you suggested is our ultimate goal that we intentionally focused on the analysis of the diurnal variation pattern of  $PM_{2.5}$ . However, the observed salient data gaps in using our retrieved  $PM_{2.5}$  time series became a big obstacle and this is also the motivation of the development of this gap filling method. In the next step, we will attempt to extract the general pattern of  $PM_{2.5}$  diurnal variation in space and time using the gap filled time series and then use such general patterns to better deal with data gaps present in future data records. In short, your insightful suggestion provides us new perspective to use  $PM_{2.5}$  diurnal variation pattern to better deal with  $PM_{2.5}$  data gaps in the future. Also, we have discussed this perspective in our revised manuscript.

(b) The PM2.5 diurnal variation could be related to some specific meteorological factors as well as their diurnal evolution. Is it possible that the diurnal variation of specific meteorological factors be considered within the authors recommended missing value filling method?

**Reply:** Thanks for your insightful comments. Incorporating the diurnal cycle of some related factors such as the mixing layer depth would be a big plus in advancing the estimation of the  $PM_{2.5}$  missing values. Nevertheless, such an endeavor is still subject to the following two constraints: (1) the lack of high temporal resolution data (e.g., hourly mixing layer depth) and (2) the relationship between  $PM_{2.5}$  and the related factor (that is, to what extent the diurnal variation pattern of PM2.5 can be explained by the diurnal cycle of the related factor). To figure out the mechanism, more explicit model simulations are anticipated, which is out of the scope of our current study. However, it deserves more in-depth analysis in the future. Such an endeavor has been discussed in the revised manuscript to provide new envisions to the improvement of this gap filling method.

(c) What is the applicability of the method? Especially for the different spatial distribution of the air quality monitoring stations which are condense over eastern China but sparse over western part of the country.

**Reply:** Good point! The question you mentioned does matter the accuracy of the proposed method and we have discussed this issue in our revised manuscript in terms of the impact of number of neighboring stations on the prediction accuracy (Fig.9b). For stations with at least one neighboring monitor (like in the west of China), our method is still able to recover the missing value with good accuracy (R>0.7). This lies in the principle that both spatial and temporal neighborhood information are used to reconstruct the diurnal cycle of PM2.5. Such effect is also corroborated by our most recent paper (Bai et al., *Environmental Pollution*, 2019, doi: 10.1016/j.envpol.2019.113047) that PM2.5 has a good autocorrelation in adjacent space and time domain. The prediction accuracy could be relatively poor for those isolated stations (no neighboring station) given the lack of spatial neighborhood information, and such effect might be mitigated by incorporating other related factors as you suggested in the future. We have discussed this issue in this revision to bridge the readership gap.

**Figure 9.** Impacts of the number of missing values present in hourly PM2.5 records for every 24-h (a) and the total number of neighboring stations within 100 km (b) on the performance of the proposed gap filling method. The error bars denote one standard deviation of each value from the mean on each side.

**References:**

- Bai, K., Li, K., Chang, N.-B., Gao, W., 2019. Advancing the prediction accuracy of satellite-based PM2.5 concentration mapping: A perspective of data mining through in situ PM2.5 measurements. *Environ. Pollut.* 254, 113047. https://doi.org/10.1016/j.envpol.2019.113047
- (d) In the manuscript, the authors made cross validation for missing value filling for several hours, is it possible that there are missing value for a specific station for one day or several days? If this situation happens, how about the performance of the authors recommended method to make missing value filling?

**Reply:** In the current manuscript, we only deal with the days with missing values no more than 20 within 24 hours since the missing values are recovered by projecting the reconstructed diurnal cycle of PM2.5 to the level of valid measurements on a specific day. For the situation with data missing for a whole day or several consecutive days, we did not recover the data given the lack of essential reference data values. Although there exists a possible way that the diurnal cycle of other related factors could be used, data amplitudes on different days may still differ from each other even in the presence of similar diurnal cycle

pattern, and this is also the reason why we need to have several valid measurements for that specific day. This issue has been discussed in the revised manuscript to bridge the readership gap. Again, we highly appreciate your insightful comments in helping improve the quality of this manuscript.

**Some specific comments are also listed below for the authors.**

 ine 60, "data cleaning processes", consider using more accurate wording to describe what the authors want to mention.

Reply: Per your kind suggestion, it has been reworded to "how data gaps were treated in their data exploration processes (e.g., integration and transformation)" to ease the readership.

2. Lines 70-71, it is better to directly give the disadvantages of "approaches of ignoring missing values or excluding records on days with missing values", rather than arbitrarily comment these approaches as "unreasonable".

Reply: Per your kind suggestion, the disadvantages have been clearly stated in the revised manuscript as: "Nevertheless, such a treatment on missingness (i.e., ignoring missing values or excluding records on days with missingness) would either introduce new bias to the aggregated PM2.5 record or make the original PM2.5 time series temporally discontinuous, especially when missingness occurs at some specific times (e.g., during severe pollution episodes)."

3. Table 1, the lines for the references are not quite clear, it is difficult to find which reference is corresponding to which method.

Reply: Thanks for pointing it out. We have enlarged the height of each row to make them more distinguishable.

4. Line 152, "*m* was defined as the number of stations within 100 km of the target station", as the authors mentioned about the "significant heterogeneity" of the PM2.5 data, is the setting of "100 km" improperly greater in this context? PM2.5 concentration can vary largely even within a small area. Moreover, the air quality monitoring stations are densely distributed over eastern China but sparsely over western part of China. Is there any special consideration should be taken on this issue?

Reply: Thanks for your insightful comments. We are aware of the fact that m and n are two critical factors associated with the performance of the proposed gap filling method since it determines the size of spatial and temporal neighborhood used for the reconstruction of the diurnal cycle of PM25. In the current method, two empirically-determined invariant spatial and temporal window sizes of 100 km and 14 nearterm days were used, but these two numbers have little effect on the final prediction accuracy of missing values. This is because these two numbers are simply used as a threshold to limit the number of samples to avoid the usage of all available data. Our recent study published in Environmental Pollution has revealed a proper spatial and temporal window size of autocorrelation of 50 km and 3-day in eastern China for PM2.5. Therefore, a window size of 100 km and 14-day suffices to include adequate number of candidate samples in space and time for the reconstruction of PM2.5 diurnal cycle. Most critically, the neighboring data confined to these two numbers are not directly used to reconstruct the diurnal cycle of  $PM_{2,5}$ ; rather, we have proposed a set of constraints to pinpoint those similar samples from the whole dataset determined by 100 km and 14-day for the subsequent diurnal cycle reconstruction. Finally, only those samples have similar diurnal variation pattern will be used for the diurnal cycle reconstruction. We have clearly stated this in section 3, please refer to the second procedure (construct a compact PM2.5 neighborhood field) on page 8 (Line 156-160) for more detail.

5. The day-to-day PM2.5 diurnal variation could vary largely, which depends on whether it is a clean day or a severe polluted day, as well as the various weather conditions. The authors also mentioned this in Lines 302-304. While the method the authors suggested only considers the diurnal variation of one week before and one week after the data missing day to be filled. Is it possible any variety in the diurnal variation of PM2.5 can be considered in the recommended method? Also, more detailed classification and establishment of the typical patterns of PM2.5 diurnal variation and adequate consideration of this issue could be very helpful to improve the data filling method suggested.

Reply: We appreciate your constructive suggestion. Same as the above question, the temporal window used here would not significantly affect our results since it is simply a cutoff value (large enough to include adequate samples) to limit the number of samples for the subsequent analysis. A compact neighborhood is further generated for the reconstruction of PM2.5 diurnal cycle by only including similar

samples rather than all the data samples. The classification of the typical patterns of  $PM_{2.5}$  diurnal variation is quite a smart suggestion and we will try to account for this effect in the further to improve the current method. We have envisioned this perspective in the discussion section to broaden the possible improvement of the current method. Again, thanks for your insightful suggestion.

6. Figure 3, it is a little difficult to understand the variables illustrated. The result presented in each panel of the figure seems not match with the caption. The name of the x axis in Figure 3f could be better as "hour".

Reply: Thanks for pointing these typos. Following your suggestions, we have revised the caption to match the figure. The name of the x axis in Figure 3f has been revised to "hour" per your suggestion.

7. Figure 4a, the 50th percentile of the mean relative differences generally remains constant around zero, does this mean that the 50th percentile is subjective of less influence from missing values?

Reply: The 50th percentile of the mean relative differences around zero just reveals the fact that missing values would result in random bias (half below zero and half above zero) to  $PM_{2.5}$  daily averages. We have explained this effect in the manuscript in section 4.2.2 (Line 282-286).

 Figure 6, the reconstructed diurnal PM2.5 variation seems to be a smoothed average of the observations near the interested station within a week before and after the interested day, it cannot reconstruct any particular variation of PM2.5 such as those at 19:00 local time in Figure 6e and at 08:00-09:00 local time in Figure 6f.

Reply: Yes, the reconstructed diurnal cycle of  $PM_{2.5}$  is relatively smooth compared with the actual observations and thus some small variations cannot be fully recovered. This lies in the fact that the diurnal cycle of  $PM_{2.5}$  is reconstructed from the spatial and temporal neighborhood using the EOF method and hence it mainly captures the dominant variation mode. We have clearly explained this defect in our manuscript in Lines 336-340.

9. Lines 409-411, because of the "significant heterogeneity" of the PM2.5 spatial distribution, how about the spatial distribution of the diurnal pattern of PM2.5 variation? Is it practical to consider the variability of PM2.5 at the stations 100 km away to fill missing value of PM2.5?

Reply: Thanks for your constructive comments. Yet the spatial distribution of the diurnal pattern of  $PM_{2.5}$  variation in China has not been examined due to the discontinuous hourly  $PM_{2.5}$  observations, we will investigate the diurnal pattern of  $PM_{2.5}$  variation in China soon per your suggestion and try to identify the typical diurnal variation pattern to improve the current method. In our current method, we did not consider to use  $PM_{2.5}$  data measured 100 km away for gap filling though there might exist similar variation patterns. This lies in the first principle of geography that data from closer stations are more similar than those distant away. On the other hand, the final prediction accuracy is not sensitive to the spatial window size if it is large enough to include three neighboring stations (Figure 9b).

10. Do Figure 10a and 10b reflect the same information from different perspectives? Is it possible just keep one figure to discuss the issue?

Reply: Not exactly. Actually, Figure 10a indicates the total number of missingness (percentage with respect to the total number of record) have been filled at each station whereas Figure 10b shows the number of days with missingness removed. As shown in Figure 10a, the removed total number of missingness seems to be low compared with the total number of samples. Nevertheless, Figure 10b indicates the percentage of how many days are without missingness after gap filling.

11. Lines 414-422 and Figure 10, have the authors done data filling for all the available PM2.5 data over China with the recommended method? Is the evaluation presented here are based on data filling for the whole dataset of PM2.5 available?

Reply: Yes, we have performed gap filling for each site-specific  $PM_{2.5}$  record in China and the results shown in Figure 10 are based on the gap-filled data set. As indicated, data gaps still persist even after gap filling and this is mainly because we did not fill the gaps for the episodes with missingness continuing for a whole day or several consecutive days. Discussions with respect to this issue has been added to fill the readership gap.

**Referee #2:**

The submitted manuscript well fits within the journal scope as it is describing a method to fill missing values in hourly PM2.5 concentrations for more than one thousand observational sites across China. Overall, the work is consistent and the method is well explained. Nevertheless, in my opinion, before publication, two points should be considered before publication

Reply: Thank you for your valuable comments and suggestions in helping improve the quality of this manuscript. The paper has been thoroughly revised according to your comments (in black), and please find the point-to-point responses (in blue) to your concerns below and refer to the revised paper for more detail.

 The authors made a sensitivity study to assess how the number of neighbour stations impact the reconstruction of PM2.5 concentration. However, it might happen that the spatial distribution of the neighbour station might influence the final result, i.e. in case of equispatially distributed or spreade. I suggest to perform a sensivity test for a couple of cases taking as metric the sum of euclidean distances using the same number of stations for the same aerosol loading.

Reply: Good point. Per your kind suggestion, we checked the potential impacts of the number of neighboring stations and their spatial structure on the prediction accuracy of missing values, which is shown in Figure R1. It can be seen that the correlation coefficient does not changes dramatically with the increase of number of neighboring stations as well as the distance between the target station and the closest station. This means that the spatial pattern of neighboring station does not influence the performance of the proposed gap filling method. This is mainly due to the implementation of an optimization process (step 2 in our method) to identify similar observations rather than using all available observations for the reconstruction of PM2.5 diurnal cycle. In other words, the final input observations only contain those with similar diurnal variation pattern to the target observation, and the distance is thus not a critical influential factor when there exist abundant samples.

Figure R1. Impacts of number of neighboring stations and their spatial structure on the prediction accuracy of missing values.

 it is missing how the measurement error is impacting the reconstruction as all the measurements are presented without error bars.

Reply: Thanks for your valuable comments. The impact of measurement error on the final accuracy of gap filling is not assessed in the current manuscript. The reasons are twofold: (1) The PM2.5 data used in this study are gauged by the state-level monitors, so the quality of the data record is assured. (2) Our gap filling method mainly get involved in the usage of empirical orthogonal function (EOF) in order to reconstruct the diurnal variation pattern of PM2.5, which would in turn cancel out the measurement errors (if any). Therefore, the measurement error would have little effect on the final results.

English should be revised as some sections are not very clear.

Reply: We have made essential corrections in this revised manuscript per your valuable suggestion.

Specific comments are available in the attached file.

Reply: Thanks for your valuable comments and suggestions. Except for the glitches and typos that have been corrected directly in our revision, the responses to several specific concerns are listed as follows.

Line 152: how those numbers (m and n) are determined? How the method accuracy changes changing those numbers?

Reply: *m* and *n* are determined by the given spatial (100 km) and temporal (7-day before and after t) window size, respectively. A cutoff value of 100 km and 7-day are used based on our recent results in which an optimal window size of 50 km and 3-day was found to attain a good autocorrelation of PM2.5 concentration in space and time, respectively (Bai et al., 2019). Here we enlarge (double) the both window sizes so as to have adequate samples for the construction of  $X_{p,t}^{m,n}$  while avoiding including all available samples, especially for those distant away. In general, these two window sizes would have little effect on the performance of the subsequent gap filling once they are large enough (at least greater than the identified optimal window sizes) to cover most similar observations nearby since a sorting scheme (step 2) will be further applied to identify observations with similar diurnal variation patterns to that of the target station. Such effect is also evidenced in Figure 9b that the prediction accuracy would not increase with the number of neighboring stations once there are more than 3 neighboring stations nearby. These more detailed discussions have been added in the revised manuscript to ease the readership.

**Line 171-175: this part should be better explained.**

Reply: This part regarding the EOF process for data gap filling has been explained by clarifying it in the context in this revision, which shows as follows:

"Reconstruct the local diurnal cycle of PM2.5: The diurnal cycle of PM2.5 at site p on date t (denoted as  $\beta_p^t$ ) was then reconstructed from the optimized PM2.5 neighborhood field  $X^k$  using EOF in an iterative process similar to the DINEOF method (Beckers and Rixen, 2003). In our DCCEOF method, the target PM2.5 time series at site p on date t (denoted as  $x_p^t$ ) were also included to constrain the reconstruction of  $\beta_p^t$ , and the whole field was then denoted as X.

$$\mathbf{X} = \{ \mathbf{x}_p^t, \mathbf{X}^k \} \tag{4}$$

In general, the EOF-based gap filling process can be outlined as follows: a) 20% of valid PM2.5 observations in X were first held out for cross validation and then these data values were treated as gaps by replacing with nulls (i.e., missing value); b) given that a small amount of missing values would not significantly influence the leading EOF mode for the original data set, we may assign a first guess (here we used the mean value of valid data on each specific date) to the data points where missing values are identified to initialize the EOF analysis; c) EOF analysis was performed on the previously generated background field (that is, X with gaps are filled with daily mean and denoted as  $\langle X \rangle$ ) in a form of singular value decomposition (SVD) and then data values at value-missing points were replaced by the reconstructed values using the first EOF mode. These processes can be expressed as:

$$[U, S, V] = svd(\langle X \rangle) \tag{5}$$

$$X' = u_1 * s_1 * v_1 \tag{6}$$

where  $\langle X \rangle$  denotes the initial matrix in which the missing values were filled with daily means. U, S, and V are three matrices derived from SVD while  $u_1$ ,  $s_1$ , and  $v_1$  denote the SVD components in the first EOF mode."

**Line 355: "largely from that of neighboring stations at the same time", how do you deal with this problem?**

Reply: The proposed DCCEOF method is unable to deal with such issue once the diurnal variation pattern of neighbors differs largely from that of the target station. We have clearly stated this defect in our revised manuscript.

Line 360: how about the instrument precision?

Reply: The precision of PM2.5 records have been introduced in section 4.1.

Line 409: how about the spatial distribution of the stations? How this impacts on final result? "The experimental results suggest that three neighboring stations within 100 km", does matter the their mutual location?

Reply: The topotaxy effect between these neighboring stations is not considered in our current method since we only accounted for the relative similarity between their diurnal variation patterns rather than their locations. In other words, whether the PM2.5 observation measured at one station will be applied for gap filling does not depend on its location (see Figure S1); Rather, we only took the similarity of PM2.5 observations between the target station and neighboring stations as a measure to select similar observations for the subsequent diurnal cycle reconstruction. Discussions related to this issue has been added in the revised manuscript to bridge the readership gap.

**Filling the gaps of in-situ hourly PM2.5 concentration data with the aid of empirical orthogonal function constrained by diurnal cycles**

Kaixu Bai1,2, Ke Li2, Jianping Guo3, Yuanjian Yang4,5, Ni-Bin Chang6

1Key Laboratory of Geographic Information Science (Ministry of Education), East China Normal University, Shanghai 200241, China

2School of Geographic Sciences, East China Normal University, Shanghai 200241, China

3State Key Laboratory of Severe Weather, Chinese Academy of Meteorological Sciences, Beijing 100081, China

4School of Atmospheric Physics, Nanjing University of Information Science & Technology, Nanjing, China

5Institute of Environment, Energy and Sustainability, The Chinese University of Hong Kong, Hong Kong, China

6Department of Civil, Environmental, and Construction Engineering, University of Central Florida, Orlando, FL 32816, USA

Correspondence to: Dr./Prof. Jianping Guo (jpguocams@gmail.com)

Abstract. Data gaps frequently emerge in our retrieved in-situ hourly air quality data records. In this study, we propose a novel gap filling method called the diurnal cycle constrained empirical orthogonal function (DCCEOF) to fill in data gaps for the improvement of data completeness. The hourly PM2.5 concentration data retrieved from the China national air quality monitoring network is used here as a demonstration. Generally, the DCCEOF method works in a principle of calibrating the diurnal cycle of PM2.5 concentration that is reconstructed from discrete PM2.5 neighborhood fields in space and time to the level of valid PM2.5 concentration observed at adjacent times. Prior to gap filling, the data completeness and the impact of data gaps in hourly PM2.5 concentration record on daily averages, were examined. The statistical analysis indicates a high frequency of data gaps in our retrieved hourly PM2.5 record, with PM2.5 concentration measured on about 40% of days subject to data gaps. On the other hand, these data gaps could introduce significant bias to daily averages of PM2.5 concentration, especially during clean episodes as larger biases would be introduced to PM2.5 daily averages during clean days than polluted days even in the presence of same number of missingness. The cross-validation results indicate that the DCCEOF method has a good prediction accuracy, particularly in predicting daily peaks and/or minima that cannot be restored by the conventional spline interpolation approach, given the consideration of local diurnal variation pattern of PM2.5 in our method. A practical application of the DCCEOF method to the retrieved hourly PM2.5 record in China during 2014 to 2019 yields a significant improvement of the data completeness, with the frequency of days with data gaps reduced from 42.6% to 5.7%. In general, the results in this study have well demonstrated the performance and application potential of DCCEOF in handling data gaps in time series of geophysical parameters with significant diurnal variability, and this method can be easily applied to other data sets with similar barriers because of its self-consistent capability.

**Deleted:** are ... requently observed ... merge in the ... ur retrieved in-situ hourly PM25 mass concentration... ir quality data recordss measured from the China national air quality monitoring network... this study, we proposed... a novel gap filling method called the diurnal cycle constrained empirical orthogonal function (DCCEOF) to fill in data gaps ... for the improvement of data completenesspresent in hourly PM25 concentration records [11]

**(Formatted: Subscript**

Deleted: This ... he DCCEOF method mainly ... orks in a principle of calibrates ... alibrating the diurnal cycle of PM2.5 concentration that is reconstructed from discrete PM2 s neighborhood fields in space space and time to the level of valid PM2.5 data values...oncentration observed at adjacent times. Prior to gap filling, the data completeness and possible ... he impacts ... of varied number of ... ata gaps in the time series of ... ourly PM2.5 concentration record on PM2.5 ... aily averagess...were ...ere examined via sensitivity experiments... The statistical analysis indicates a high frequency of data gaps in our retrieved hourly PM2s record, with PM2s concentration measured on about 40% of days The results showed that PM2.5 data suffered from...ubject to data the ... aps on about 40% of days, indicating a high frequency of missing data in the hourly PM2.5 records... On the other hand, tT...ese datase...gaps could introduce significant bias to daily -...verages ofd ... PM2.5 concentration. [2]

**Formatted: Subscript**

Deleted: Particularly, given...ven in the presence of the ...ame number of gaps...issingness, larger biases would be introduced to daily-averaged PM2.5 during clean days than polluted days... The cross-validation results indicate that the predicted missing values from ... he DCCEOF method has a good prediction accuracy with the consideration of the local diurnal phases of PM2.5 are more accurate and reasonable than those from the conventional spline interpolation approach ... especially for ... articularly in the reconstruction ... redicting of ... aily peaks and/or minima that cannot be restored by the conventional spline interpolation approachlatter method... given the consideration of local diurnal variation pattern of PM2.5 in our method. To fill the gaps ... practical application of the DCCEOF method toin...the retrieved hourly PM2.5 records...across in China during 2014 to 2019, as a practical application, the DCCEOF method can be able to ... yields a significant improvement of the data completeness, with reduce ... hethe ... averaged ... requency of days with missingness ... ata gaps reduced from 42.6% to 5.7%. In general, the present work implies...he results in this study have well demonstrated that ... he performance and application potential of DCCEOF method is realistic and robust to be able to ... n handle handling the missingness issues...ata gaps in time series of geophysical parameters with significant diurnal variability, and this method can be easily applied and can be expectably applied ...toin[3]

**1** Introduction**

A large variety of ground-based monitoring networks have been established worldwide to provide accurate measurements on various aspects of the atmospheric environment (Lolli and Di Girolamo, 2015), Many of these in-situ measurements, however, suffer from data losses due to various unexpected reasons, e.g., instrumental malfunction, interruption of power supply, and internet outage, thus resulting in salient data gaps in the archived data records. Undoubtedly, these gaps significantly impair the data qualities and the exploration of these valuable data sources. Therefore, filling data gaps present in such datasets is critical and of great value to facilitating the broad application of these in-situ measurements.

Confronted with frequently occurring severe haze pollution events, China started to establish the national ambient air quality monitoring network since 2012 by extending the range of the previous sparsely distributed monitoring network to cover most major Chinese cities. To date, more than 1,600 state-level stations are routinely operated to measure concentrations of six essential air pollutants (i.e., PM10, PM2.5, O3, NO2, SO2, CO) on an hourly basis (Guo et al., 2017; Li et al., 2017a). These in-situ measurements are publicly released online via the China National Environment Monitoring Centre (CNEMC) in near real-time as of 2013 but without providing any direct data download interface. Consequently, users oftentimes use an automated software program (often known as a "web crawler") to retrieve these valuable data sources from the CNEMC website. Such an endeavour helps users to acquire hourly air quality data more efficiently, and the retrieved data record, taken PM2.5 mass concentration data as an example, have been widely used as a critical data source in many haze related studies (Gao et al., 2018; Miao et al., 2018; Bai et al., 2019a, 2019b; Zhang et al., 2019).

Although these PM2.5 concentration data have been extensively used, how data gaps were treated in the data exploration process (e.g., data integration and data transformation), especially for those using daily or monthly averaged PM2.5 data set (e.g., Guo et al., 2009; Miao et al., 2018; Ye et al., 2018; Zhang et al., 2018; Yang et al., 2019a), is oftentimes unclear. Since ignoring missing values would undoubtedly introduce biases into the final results (Bondon, 2005; Larose et al., 2019), some studies attempted to perform data analysis on a relatively long time scale to mitigate the impacts of data gaps by integrating hourly records into monthly resolution (e.g., Bai et al., 2019b; Zhang et al., 2019). On the other hand, many previous studies preferred to exclude records on days subject to a certain degree of missing values

| (      | Deleted: such as the Aerosol Robotic Network (AERONET) for aerosol properties |
|--------|-------------------------------------------------------------------------------|
| (      | Deleted: accidents                                                            |
| (      | Deleted: either on monitoring stations or user's end                          |
| ·····( | Deleted: thereby                                                              |
|        |                                                                               |
| (      | Deleted: their                                                                |
| ····(  | Deleted: applications                                                         |
| $\geq$ | Deleted: the                                                                  |
| (      | Deleted: ed                                                                   |

**Deleted: utilize**

| Deleted: . The retrieved hourly mass concentration record, taking PM2.5 for instance,         Deleted: has         Deleted: has         Deleted: related to haze pollutions,         Deleted: because of its good accuracy and high temporal resolution as well as its national-scale coverage         Deleted: from this dataset         Deleted: in many PM2-s-related studies         Deleted: during the         Deleted: cleaning         Deleted: es         Deleted: particularly |                                         |                                                                                         |
|------------------------------------------------------------------------------------------------------------------------------------------------------------------------------------------------------------------------------------------------------------------------------------------------------------------------------------------------------------------------------------------------------------------------------------------------------------------------------------------|-----------------------------------------|-----------------------------------------------------------------------------------------|
| Deleted: has         Deleted: related to haze pollutions,         Deleted: because of its good accuracy and high temporal resolution as well as its national-scale coverage         Deleted: from this dataset         Deleted: in many PM2.5-related studies         Deleted: the method of treating         Deleted: during the         Deleted:         Deleted: cleaning         Deleted: es         Deleted: particularly                                                           | Deleted: .
PM 2.5 for ins | The retrieved hourly mass concentration record, taking stance,                          |
| Deleted: related to haze pollutions,         Deleted: because of its good accuracy and high temporal resolution as well as its national-scale coverage         Deleted: from this dataset         Deleted: in many PM2-3-related studies         Deleted: the method of treating         Deleted: during the         Deleted: cleaning         Deleted: es         Deleted: particularly         Deleted: of                                                                             | Deleted: h                              | as                                                                                      |
| Deleted: because of its good accuracy and high temporal resolution as well as its national-scale coverage         Deleted: from this dataset         Deleted: in many PM2.5-related studies         Deleted: the method of treating         Deleted: during the         Deleted: cleaning         Deleted: es         Deleted: particularly         Deleted: of                                                                                                                          | Deleted: 1                              | related to haze pollutions,                                                             |
| Deleted: from this dataset         Deleted: in many PM2.5-related studies         Deleted: the method of treating         Deleted: during the         Deleted:         Deleted:         Deleted: cleaning         Deleted: es         Deleted: particularly         Deleted: of                                                                                                                                                                                                          | Deleted: I
resolution as             | because of its good accuracy and high temporal
s well as its national-scale coverage |
| Deleted: in many PM2-5-related studies         Deleted: the method of treating         Deleted: during the         Deleted:         Deleted: cleaning         Deleted: es         Deleted: particularly         Deleted: of                                                                                                                                                                                                                                                              | Deleted: f                              | rom this dataset                                                                        |
| Deleted: the method of treating         Deleted:         Deleted:         Deleted:         Deleted: cleaning         Deleted: es         Deleted: particularly         Deleted: of                                                                                                                                                                                                                                                                                                       | Deleted: i                              | in many PM2.5-related studies                                                           |
| Deleted:         Deleted:         Deleted: cleaning         Deleted: es         Deleted: particularly         Deleted: of                                                                                                                                                                                                                                                                                                                                                                | Deleted: t                              | he method of treating                                                                   |
| Deleted:         Deleted: cleaning         Deleted: cs         Deleted: particularly         Deleted: of                                                                                                                                                                                                                                                                                                                                                                                 | Deleted: d                              | luring the                                                                              |
| Deleted: cleaning
Deleted: of                                                                                                                                                                                                                                                                                                                                                                                                                 | Deleted:                                |                                                                                         |
| Deleted: es
Deleted: of                                                                                                                                                                                                                                                                                                                                                                                                                                      | Deleted: c                              | leaning                                                                                 |
| Deleted: particularly Deleted: of                                                                                                                                                                                                                                                                                                                                                                                                                                                        | Deleted: e                              | s                                                                                       |
| Deleted: of                                                                                                                                                                                                                                                                                                                                                                                                                                                                              | Deleted: p                              | varticularly                                                                            |
|                                                                                                                                                                                                                                                                                                                                                                                                                                                                                          | Deleted: o                              | f                                                                                       |
| Deleted: with                                                                                                                                                                                                                                                                                                                                                                                                                                                                            | Deleted: v                              | vith                                                                                    |

(e.g., no more than 6 missing values within 24-h) from their analysis (e.g., van Donkelaar et al., 2016; Li et al., 2017; Huang et al., 2018; Manning et al., 2018; Shen et al., 2018; Bai et al., 2019a; Zhang et al., 2019). Nevertheless, such a treatment on data gaps (e.g., ignoring missing values or excluding records on / days with missingness) would either introduce new bias to the aggregated data record or make the original / PM2.5 time series temporally discontinuous.

Since a non-gap  $PM_{2.5}$  record is essential to  $PM_{2.5}$  related haze control and environmental health risk assessment, filling data gaps presented in hourly  $PM_{2.5}$  record is thus of critical importance. Although / there exists versatile gap filling methods (e.g., Beckers and Rixen, 2003; Taylor et al., 2013; Chang et al., 2015; Dray and Josse, 2015; Gerber et al., 2018), most of them fail to properly restore missingness in /  $PM_{2.5}$  time series with high temporal resolution (e.g., hourly). In general, the conventional methods are / offentimes incapable of restoring  $PM_{2.5}$  daily peaks and/or minima since a priori knowledge of the diurnal variation pattern of  $PM_{2.5}$  is always required as  $PM_{2.5}$  mass concentration, varies significantly in space and / time due to heterogeneous local emissions and atmospheric conditions (Guo et al., 2017; Lennartson et al., 2018; Shi et al., 2018). A similar barrier also applies for many other datasets which are sampled at high temporal resolution.

In this study, we propose, a novel gap filling method termed as DCCEOF (that is, the diurnal cycle constrained empirical orthogonal function) to better handle data gaps present in time series with marked variability in space and time, by taking the diurnal variation pattern as a critical constraint in missing value prediction. To our knowledge, none of the existing gap filling methods have accounted for the diurnal variation pattern of the given data in their missing value reconstruction schemes, and hence the predicted values from these methods, are prone to large bias. As an illustration, the hourly PM2.5 concentration record retrieved from CNEMC during the time period of 2014 to 2019 is applied here to demonstrate the efficacy and accuracy of the proposed DCCEOF method. Science questions to be answered by this study include: (1) how about the data completeness of the Chinese in situ PM2.5 record? (2) how much uncertainties can be introduced to PM2.5 draily averages by missing values? (3) is it feasible / to reconstruct the Jocal diurnal variation pattern of PM2.5 from discrete observations in the neighborhood? / and (4) are missing values restored by DCCEOF reliable?

**Deleted:** Although ... evertheless, such a treatment on data gaps (e.g., ignoring missing values or excluding records on days with missingness) the exclusion of records with missingness could avoid biased results to some extent, such a data treatment ...ould either introduce new bias to the aggregated data record or make the original  $PM_{25}$  time series temporally discontinuous. Therefore, approaches of ignoring missing values or excluding records on days with missing values are unreasonable. ... [4]

00089.1","ISSN":"08948755","abstract":"Empirical orthogonal function (EOF) analysis is commonly used in the climate sciences and elsewhere to describe, reconstruct, and predict highly dimensional data fields. When data contain a high percentage of missing values (i.e., gappy), alternate approaches must be used in order to correctly derive EOFs. The aims of this paper are to assess the accuracy of several EOF approaches in the reconstruction and prediction of gappy data fields, using the Galapagos Archipelago as a case study example. EOF approaches included least squares estimation via a covariance matrix decomposition [least squares EOF (LSEOF)], data interpolating empirical orthogonal functions (DINEOF), and a novel approach called recursively subtracted empirical orthogonal functions (RSEOF). Model-derived data of historical surface chlorophyll-a concentrations and sea surface temperature, combined with a mask of gaps from historical remote sensing estimates, allowed for the creation of true and observed fields by which to gauge the performance of EOF approaches. Only DINEOF and RSEOF were found to be appropriate for gappy data reconstruction and prediction. DINEOF proved to be the superior approach in terms of accuracy, especially for noisy data with a high estimation error, although RSEOF may be preferred for larger data fields because of its relatively faster computation time. © 2013 American Meteorological Society.","author":[{"dropping particle":"","family":"Taylor","given":"Marc H.","non-droppingparticle":"" ,"parse-names":false,"suffix":""},{"droppingparticle":"" ,"family":"Losch","given":"Martin","non-dropping particle":"" ,"parse-names":false,"suffix":""},{"droppingparticle":"" "family":"Wenzel","given":"Manfred","non-droppingparticle":"" "parse-names":false,"suffix":""},{"droppingnarticle":"" "family":"Schröter","given":"Jens","non-droppingparticle":"","parse-names":false,"suffix":""}],"containertitle":"Journal of Climate","id":"ITEM-1","issue":"22","issued":{"date-parts":[["2013"]]},"page":"9194-.. [5]

Deleted: d...a novel practical ...ap filling method called ...ermed as DCCEOF (that is, the a ...iurnal cycle constrained empirical orthogonal function (DCCEOF... to better handle data gaps presented...in time series with marked variability in space and time, by taking the diurnal phases ...ariation pattern as a critical constraint in missing value imputation...rediction. To our knowledge, none of the existing gap filling methods have accounted for the diurnal phase variation pattern of the given dataeffect... in their missing value imputation ...construction schemes, and hence the predicted values from these methods might ...are suffer from...rone to large bias. As an illustration demonstration... the retrieved ...ourly PM2.5 concentration record retrieved from CNEMC during the time periode

**(Formatted: Subscript**

**2 Overview of existing gap filling methods**

Plenty of methods have been developed or adopted for gap filling with respect to various theoretical bases, ranging from simple replacement with surrogates (e.g., mean value) to spatiotemporal interpolation as well as complicated machine learning techniques, Generally, these methods can be classified into different groups according to different criteria. For instance, two major groups can be classified based on the number of variables (univariate versus multivariate) (Ottosen and Kumar, 2019) and theoretical basis (likelihood-based versus imputation-based) (Junger and Ponce de Leon, 2015). Table 1 summarizes a selection of popular gap filling methods to deal with missingness in geophysical data sets according to the domain specific data dependence (Gerber et al., 2018). Comparisons of the performance of these methods can also be found in other literatures, e.g., Kandasamy et al. (2013), Demirhan and Renwick (2018), Yadav and Roychoudhury (2018), and Julien and Sobrino (2019), to name a few.

Since each method is initially proposed to deal with missingness in one specific data set, adopting one method to another data set is often a challenge due to distinct features of missingness (e.g., missing at random versus missing not at random), in particular for data sets with salient spatiotemporal heterogeneity such as air pollutants time series (Junger and Ponce de Leon, 2015). PM2.5 concentration often exhibits evident diurnal variation patterns, which are primarily governed by local air pollutants emissions and regional meteorological conditions such as boundary layer height (Guo et al., 2017; Li et al., 2017; Huang et al., 2018; Liu et al., 2018; Miao et al., 2018; Yang et al., 2018, 2019b). Consequently, conventional approaches like those listed in Table 1 may partially fail in accurately predicting missing values in hourly PM2.5 time series.

In general, most available gap filling methods in Table 1 suffer from at least one of the following drawbacks: 1) partially fail for data sets with prominent gaps; 2) not self-consistent due to the requirement of supplementary data sets;3) computationally intensive (e.g., neural networks), and, most critically; 4) unable to fairly predict daily peaks and/or minima due to the Jack of essential prior knowledge of diurnal variability of monitoring targets. Given the significant heterogeneity of PM2.5 concentration in space and time (Guo et al., 2017; Manning et al., 2018), ignoring the diurnal phases of PM2.5 would result in large bias to the gap filled PM2.5 data set.

| (     | Deleted: al or                   |
|-------|----------------------------------|
| ••••( | Deleted: in addition to          |
| (     | Deleted: such as neural networks |
| ``(   | Deleted: T                       |

| Deleted:   | for              |
|------------|------------------|
| Deleted:   | value imputation |
| Deleted:   | by referring     |
| Deleted:   | 5                |
| Deleted:   | 5                |
| Deleted:   | among others     |
| Deleted:   | Given that       |
| Deleted: t | the various      |

| Deleted | : ly   |
|---------|--------|
| Deleted | phases |

| Fo  | rm | at | teo | 1: | Su | bso          | rir | bt |
|-----|----|----|-----|----|----|--------------|-----|----|
| ••• |    |    |     |    | Ju | $\nu \sim 0$ |     |    |

Table 1 Overview of several nonular gan filling methods to impute missingness in geophysical data sets

| Tabi    |                                                                                                                         | several popular gap mining methods to impute mi                                                | issingness in geophysical data sets              | • // /       | Formatted  | [10  |
|---------|-------------------------------------------------------------------------------------------------------------------------|------------------------------------------------------------------------------------------------|--------------------------------------------------|--------------|------------|------|
|         | Method                                                                                                                  | Principle or core technique                                                                    | Reference                                        |              | Formatted  | [11  |
|         | Weibull                                                                                                                 | Weibull frequency distribution manning                                                         | Nosal et al. (2000)                              |              | Formatted  | [12  |
|         | weibuli                                                                                                                 | weroun nequency distribution mapping                                                           | Nosai et al. (2000)                              |              | Formatted  | [13  |
|         | EM                                                                                                                      | Expectation-Maximization                                                                       | Junger and Ponce de Leon (2015)                  | •            | Formatted  | [14  |
|         | Interpolation                                                                                                           | Linear regression Spline NAR ARIMA ARCH                                                        | Stauch and Jarvis (2006); Neteler (20            | (0);         | Formatted  | [15  |
|         | merpenation                                                                                                             |                                                                                                | Demirhan and Renwick (2018)                      |              | Formatted  | [18  |
| ral     | Machine learning                                                                                                        | Gradient Boosting, neural networks                                                             | Körner et al. (2018); Sahin et al. (201          |              | Deleted: ; | [19  |
| iodu    | SSA                                                                                                                     | Imputation using singular spectrum analysis                                                    | Mahmoudvand and Rodrigues (2016)                 |              | Formatted  | [16  |
| Ter     | DC                                                                                                                      |                                                                                                |                                                  |              | Formatted  | [17  |
|         | DS                                                                                                                      | Conditional resampling of a temporal subset                                                    | Dembélé et al. (2019) ; Oriani et al. (20 | (91          | Deleted:   |      |
|         | TIMESAT                                                                                                                 | Savitzky–Golay filter, harmonic and asymmetric                                                 | Jönsson and Eklundh (2004)                       |              | Formatted  | [22] |
|         |                                                                                                                         | Gaussian functions                                                                             |                                                  |              | Formatted  | [20] |
|         | Hybrid method                                                                                                           | Fuzzy c-means with support vector regression and                                               | Aydilek and Arslan (2013)                        |              | Formatted  | [21] |
|         |                                                                                                                         | genetic algorithm                                                                              |                                                  |              | Formatted  | [23] |
|         | IDW                                                                                                                     | Interpolate using inverse distance weighting                                                   | Shareef et al. (2016)                            | 1            | Formatted  | [24  |
| Spatial | Kriging                                                                                                                 | Internalate neighborhoods using Kriging                                                        | Rossi et al. (1994); Zhu et al. (2015            | 5);          | Formatted  | [25  |
|         | Kriging Interpolate neighborhoods using Kriging                                                                         |                                                                                                | Singh et al. (2017)                              |              | Formatted  | [28  |
|         | NSPI / GNSPI                                                                                                            | Replace or interpolate with adjacent similar pixels                                            | Zhu et al. (2012); Chen et al. (2011)            | 4            | Deleted: 1 |      |
|         |                                                                                                                         | Iteratively decompose and reconstruct spatial and                                              | Beckers and Riven (2003): Taylor et              | *            | Formatted  | [26  |
|         | EOF / DINEOF                                                                                                            | temporal subsets using empirical orthogonal function                                           | (2013); Liu and Wang (2019)                      | A.           | Formatted  | [27] |
| al      | Mosaicing                                                                                                               | Merge numerical outputs with satellite observations                                            | Konik et al. (2019)                              | 4            | Formatted  | [30  |
| por     | gapfill                                                                                                                 | Quantile regression fitted to spatiotemporal subsets                                           | Gerber et al. (2018)                             |              | Formatted  | [29] |
| tem     | gapini                                                                                                                  | Quantine regression rited to spatiotemporal subsets                                            |                                                  |              | Formatted  | [31] |
| tio-    | STWR                                                                                                                    | Spatially and temporally weighted regression                                                   | Chen et al. (2017)                               | •            | Formatted  | [33  |
| Spa     | SMIR                                                                                                                    | Learning machine created from historical spatial and                                           | Chang et al. (2015)                              | **           | Formatted  | [32] |
|         |                                                                                                                         | temporal subsets                                                                               | g (2+++)                                         |              | Formatted  | [34  |
|         | RFRE                                                                                                                    | Learning from other information using random forest                                            | Bi et al. (2018); Chen et al. (2019)             | 1            | Formatted  | [35  |
| * SSA   | : Singular Spectrum                                                                                                     | Analysis; DS: Direct Sampling; IDW: Inverse Distance                                           | Weighting; NSPI: Neighborhood Simila             | r            | Formatted  | [36  |
| Pixel   | Pixel Interpolator; GNSPI: Geo-statistical Neighborhood Similar Pixel Interpolator; EOF: Empirical Orthogonal Function; |                                                                                                |                                                  |              |            |      |
| SMart   | Information Reconst                                                                                                     | g Empirical Orthogonal Function; STWR: Spatially and a ruction: RFRE: Random Forest Regression | remporally weighted Regression; SMIR             | :            | Formatted  | [40] |
|         | manual recollist                                                                                                        | reston, rereation rundom rotot regression                                                      |                                                  | 18586 (1944) | 8 C        |      |

**3 The DCCEOF gap filling method,**

| Given the significant heterogeneity of PM2.5 diurnal variation pattern associated with local emissions of               |  |
|-------------------------------------------------------------------------------------------------------------------------|--|
| air pollutants and atmospheric conditions, we propose to apply the local diurnal variation pattern of PM 2.5 |  |
| to constrain the reconstruction of missing values in the hourly time series of PM2.5 concentration at each              |  |
| 18                                                                                                                      |  |

| <          | [14]           |
|------------|----------------|
| Formatted  | [13]           |
| Formatted  | [14]           |
| Formatted  | [15])          |
| Formatted  | [18]           |
| Deleted: ; | [19]           |
| Formatted  | [16]           |
| Formatted  | [17]           |
| Deleted: 1 | )              |
| Formatted  |  [22]   |
| Formatted  | [20]           |
| Formatted  | [21])          |
| Formatted  | [23]           |
| Formatted  | [24]           |
| Formatted  | [25]           |
| Formatted  | [28]           |
| Deleted: 1 | )              |
| Formatted  |  [26] ) |
| Formatted  | [27]           |
| Formatted  | [30]           |
| Formatted  | [29]           |
| Formatted  | [31]           |
| Formatted  | [33]           |
| Formatted  |         |
| Formatted  | [34]           |
| Formatted  | [35]           |
| Formatted  | [36]           |
| Formatted  |         |
| Formatted  | [40]           |
| Formatted  |  [38]   |
| Formatted  | [39]           |
| Formatted  | [41])          |
| Formatted  | [42]           |
| Formatted  | [43]           |
| Formatted  | [45]           |
| Formatted  | [46]           |
| Formatted  | [44]           |
| Formatted  | [47]           |
| Formatted  | [48]           |
| Formatted  | [49]           |
| Formatted  | [50]           |
| Formatted  | [51]           |
| Formatted  | [52]           |
| Formatted  | [53]           |
| Formatted  | [54]           |
| Formatted  | [55]           |
| Formatted  | [57]           |
| Formatted  | [56]           |
| Formatted  | [58]           |
| Formatted  | [59]           |

... [9]

[8]

Formatted

Formatted Table

station. The goal is to better predict missing PM2.5 values, especially for the daily peaks and/or minima, which are poorly predicted by conventional methods due to the absence of prior knowledge of local diurnal phases of PM2.5. Figure 1 presents a schematic illustration of the proposed DCCEOF method. In general, the DCCEOF method consists of the following four primary procedures toward the filling of data gaps present in each 24-h PM2.5 time series;

---

## Author Response (AR2)

**Response to referees' comments:**

**Referee #1**

The editor may forward the following comments to the authors, but it is not necessary to be published. Thanks. I still have a little concern about the language. A proof reading and language editing would be helpful to make the paper more readable. I list some examples below.

**Response: First of all, thanks for the editor and reviewers for taking time to review our manuscript and offering helpful and constructive suggestions! We have carefully read the kind comments by all reviewers and revised the manuscript accordingly. Please see our detailed point-by-point reply below (in blue).**

On the other hand, if the authors could comment or give some directions on how any interested reader can use their suggested method, it would be great.

**Reply: Per your kind suggestion, we have revised our manuscript in an attempt to give direction on how to use our suggested method. Please refer to the revised manuscript for more details.**

1. Lines 25-28, long sentence difficult to follow, consider to revise it as "On the other hand, these data gaps could introduce significant bias to daily averages of PM2.5 concentration, especially during clean episodes when larger biases would be introduced to PM2.5 daily averages compared to the polluted days (even with the presence of same number of missingness)."

**Reply: Revised as suggested.**

2. Line 28-31, consider to revise it as "The cross-validation results indicate that our suggested DCCEOF method, with the consideration of local diurnal variation pattern of PM2.5, has a good prediction accuracy particularly in predicting daily peaks and/or minima that cannot be restored by the conventional spline interpolation approach."

**Reply: Revised as suggested.**

3. Line 45-46, grammar error.

**Reply: Corrected.**

[revised manuscript text omitted]